# Proteome-wide quantification of inositol pyrophosphate-protein interactions

Annika Richter[1,2,4], Jaime A. Isern [1,4], Max Ruwolt [1], Sarah Lampe [1,2], Abhirup Majumdar [1,2], Fan Liu [1,3], David Furkert [1,2] ✉ & Dorothea Fiedler [1,2] ✉

Inositol polyphosphates (InsPs) and inositol pyrophosphates (PP-InsPs) are highly phosphorylated signaling molecules involved in diverse cellular processes. To resolve discrete signaling events mediated by these structurally related metabolites, a mass spectrometry–based approach was developed to derive apparent binding constants on a proteome-wide scale. The method employs chemically synthesized affinity reagents for inositol hexakisphosphate ($InsP_6$) and the inositol pyrophosphates $1PP\text{-}InsP_5$, $5PP\text{-}InsP_5$, and $1,5(PP)_2\text{-}InsP_4$ ($InsP_8$). Concentration-dependent affinity enrichment combined with tandem mass tag (TMT) labeling enabled identification and quantification of ligand–protein interactions for hundreds of proteins from mammalian cell lysates. Biochemical and functional validation of selected targets demonstrated engagement with endogenous ligands. Comparison of enrichment conditions revealed a strong dependence of PP-InsP binding on $Mg^{2+}$ ions. Additionally, gene ontology analysis linked PP-InsP interactors to nuclear and nucleolar RNA processing, and subsequent analyses could identify several pyrophosphorylation sites, previously uncharacterized. In summary, these datasets provide valuable resources for exploring PP-InsP–dependent signaling pathways across biological systems.

Inositol polyphosphates (InsPs) are soluble, *myo*-inositol–based signaling molecules that are found throughout eukaryotes. A widely-studied example is inositol-1,4,5-trisphosphate ($InsP_3$), which is produced from membrane-bound phosphatidylinositol-4,5-bisphosphate ($PIP_2$) upon phospholipase C activation, and mediates calcium release from the endoplasmic reticulum[1]. Sequential phosphorylation of $InsP_3$ gives rise to more densely phosphorylated InsPs, including inositol hexakisphosphate ($InsP_6$), a molecule that can function as a structural cofactor in mammals, while serving as a major phosphate storage compound in plants[2–6].

Although InsPs are structurally more diverse than the related phosphoinositides (PIPs), they remain comparatively less studied. In recent years, however, a subgroup of InsPs - the inositol pyrophosphates (PP-InsPs)—has drawn increasing attention. These metabolites are generated through the action of inositol hexakisphosphate kinases (IP6Ks), which phosphorylate $InsP_6$ at the 5-position, and diphosphoinositol pentakisphosphate kinases (PPIP5Ks), which target the 1-position. Together, these enzymes produce 5-diphosphoinositol pentakisphosphate ($5PP\text{-}InsP_5$), 1-diphosphoinositol pentakisphosphate ($1PP\text{-}InsP_5$), and bis-1,5-diphosphoinositol tetrakisphosphate ($1,5(PP)_2\text{-}InsP_4$) (Fig. 1a)[7–13]. The dephosphorylation of these densely phosphorylated metabolites is driven primarily by diphosphoinositol polyphosphate phosphatases (DIPPs).

A putative role in cellular signaling has been attributed to the PP-InsPs for a long time, as these molecules are rapidly turned

[1]Leibniz-Forschungsinstitut für Molekulare Pharmakologie (FMP), Robert-Rössle-Straße 10, Berlin, Germany. [2]Institut für Chemie, Humboldt-Universität zu Berlin, Brook-Taylor-Straße 2, Berlin, Germany. [3]Charité - Universitätsmedizin Berlin, Charitéplatz 1, Berlin, Germany. [4]These authors contributed equally: Annika Richter, Jaime A. Isern. ✉e-mail: david.furkert@novartis.com; fiedler@fmp-berlin.de

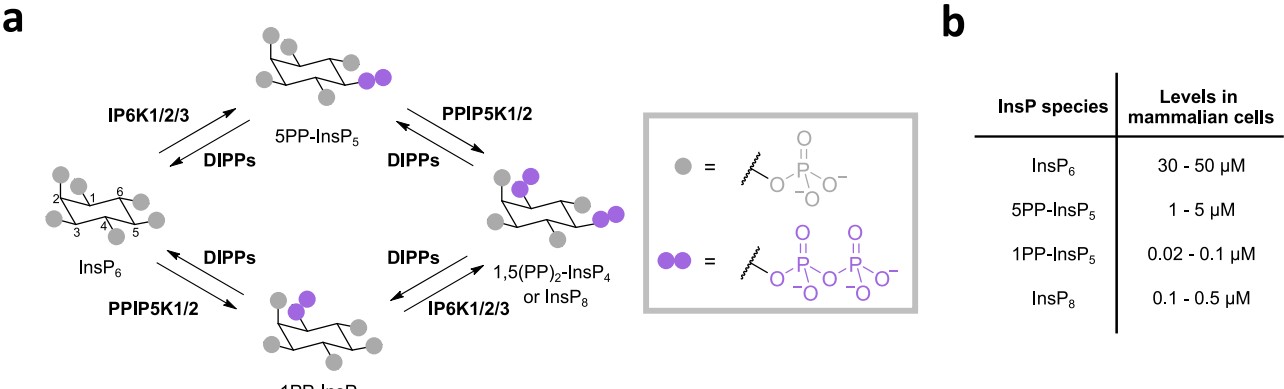

**Fig. 1 | Inositol pyrophosphate biosynthesis and estimated levels in mammalian cells. a** Pathway for the biosynthesis and turnover of inositol pyrophosphates: Inositol pyrophosphates are synthesized from $InsP_6$ (inositol hexakisphosphate), which can be phosphorylated by IP6K1/2/3 to generate $5PP\text{-}InsP_5$ (5-diphosphoinositol pentakisphosphate) or by PPIP5K1/2 to produce $1PP\text{-}InsP_5$ (1-diphosphoinositol pentakisphosphate). These molecules can be further phosphorylated to yield $1,5(PP)_2\text{-}InsP_4$ (bis-1,5-diphosphoinositol tetrakisphosphate,

also known as $InsP_8$), with IP6Ks converting $1PP\text{-}InsP_5$ and PPIP5Ks converting $5PP\text{-}InsP_5$. Dephosphorylation of these metabolites is predominantly mediated by diphosphoinositol polyphosphate phosphohydrolases, the DIPPs (DIPP1, DIPP2, DIPP3α, and DIPP3β). The main fluxes for the synthesis and turnover of PP-InsPs can differ between organisms and cell lines[9,81]. **b** Concentration ranges of (PP)-InsPs detected in a variety of human cell lines[16–19].

over in cells (up to ten times per hour)[14,15]. However, current analytical methods are not capable of resolving the detection and quantification of InsPs/PP-InsPs with subcellular resolution; therefore, many questions regarding local turnover and availability for signaling purposes have remained unanswered. Within cells, $InsP_6$/PP-InsP levels have been reported to range from nanomolar to micromolar concentrations (Fig. 1b)[16–19]. Among the PP-InsPs, $5PP\text{-}InsP_5$ is the most abundant species in most cell lines, with concentrations typically ranging between 1–5 μM.

Over the past years, the signaling functions of PP-InsPs have mainly been investigated using genetic methods, as well as pharmacological tools targeting IP6Ks[10,20]. These kinases, and in several cases by proxy the molecule $5PP\text{-}InsP_5$, have been associated with insulin secretion[21,22], focal adhesion dynamics[23,24], and apoptosis[25]. In these examples, $5PP\text{-}InsP_5$ accesses different modes of action for signal transduction, including competition for phospholipid-binding domains[26]. In addition, $5PP\text{-}InsP_5$ can transmit signals by transferring the β-phosphoryl group onto pre-phosphorylated proteins in a process termed protein pyrophosphorylation[27–31]. This unusual protein modification was demonstrated, for example, to regulate protein localization and protein degradation[32–34]. Compared to the functions of $5PP\text{-}InsP_5$, relatively little is known about the closely related messengers $1PP\text{-}InsP_5$ and $1,5(PP)_2\text{-}InsP_4$. Nevertheless, recent analyses of cell lines lacking PPIP5Ks, and consequently $1,5(PP)_2\text{-}InsP_4$, have sparked interest because these cell lines exhibit a growth-inhibited phenotype and a hypermetabolic state[35,36].

Deciphering the concrete signaling functions of individual PP-InsPs remains challenging. Genetic or pharmacologic perturbation of IP6Ks not only reduces $5PP\text{-}InsP_5$ levels but also simultaneously diminishes $1,5(PP)_2\text{-}InsP_4$ levels. Conversely, deletion of PPIP5Ks depletes $1,5(PP)_2\text{-}InsP_4$ but concomitantly increases the cellular amounts of $5PP\text{-}InsP_5$[36]. Therefore, phenotypic observations must be complemented by biochemical and/or biophysical data to assign function to a specific PP-InsP molecule. For example, a recent study demonstrated that the xenotropic and polytropic retrovirus receptor 1 (XPR1)—a protein that binds to $InsP_6$, $5PP\text{-}InsP_5$, and $1,5(PP)_2\text{-}InsP_4$—is regulated predominantly by the latter[37–39]. The in vitro binding affinities did not differ drastically but were found to be highest for $1,5(PP)_2\text{-}InsP_4$[40,41]. Subsequent NMR experiments revealed differential conformational plasticity of the different protein-ligand complexes, where $1,5(PP)_2\text{-}InsP_4$ engages in the most dynamic interaction mode[42].

These findings highlight the need for systematic determination of $InsP_6$/PP-InsP protein-binding affinities to elucidate which proteins are potentially regulated by which member of the PP-InsP family. Thus far, only purified proteins have been used to characterize PP-InsP–protein binding affinities in vitro, using methods such as isothermal titration calorimetry (ITC), surface plasmon resonance spectroscopy, and microscale thermophoresis[24,42,43]. These measurements can provide precise information, but cannot be used for large-scale analyses. Recently, a mass-spectrometric method for the global determination of apparent binding affinities of immobilized small-molecule ligands was reported[44–46]. Based on previous affinity enrichment experiments established in our group, we sought to implement a mass-spectrometric approach to quantify the proteome-wide interactions of $InsP_6$, $5PP\text{-}InsP_5$, $1PP\text{-}InsP_5$, and $1,5(PP)_2\text{-}InsP_4$.

Here, we report the synthesis of affinity reagents for $5PP\text{-}InsP_5$, $1PP\text{-}InsP_5$, and $1,5(PP)_2\text{-}InsP_4$, in which the labile pyrophosphate groups have been replaced by non-hydrolyzable bisphosphonate (PCP) moieties. These reagents were applied over a wide concentration range and, in combination with tandem mass tag (TMT) labeling, allowed us to extract apparent binding constants ($K_D^{app}$) for these ligands under different conditions on a proteome-wide scale. The presence of $Mg^{2+}$ ions notably influenced the protein binding profiles, underscoring the importance of metal ion coordination in PP-InsP recognition. More than 600 protein interactors with $K_D^{app} < 5$ μM were identified in the nuclear fraction for $5PP\text{-}InsP_5$ and $1,5(PP)_2\text{-}InsP_4$, including numerous components involved in RNA metabolism and ribosome biogenesis. Furthermore, based on the affinity-enrichment data, previously unknown pyrophosphoproteins could be identified. Beyond generating a quantitative map of PP-InsP–protein interactions, this study establishes a generalizable chemoproteomics approach to systematically probe PP-InsP signaling, thereby providing a foundation for uncovering how these molecules orchestrate cellular processes in different biological contexts in the future.

## Results
### Design and synthesis of biotinylated inositol pyrophosphate analogs
While $5PP\text{-}InsP_5$ appears to be the most abundant PP-InsP in many mammalian cell lines[7,17], a few reports have invoked a signaling role for the closely related molecule $1PP\text{-}InsP_5$[47]. In addition, several recent studies have substantiated a unique signaling function for $1,5(PP)_2\text{-}InsP_4$[36,37,48]. Building on our previously developed PCP-InsP affinity

reagents[49,50], we wanted to expand the set of reagents to include probes for 1PP-InsP$_5$ and 1,5(PP)$_2$-InsP$_4$. Because the modification of a phosphoryl group with a linker can influence the protein interactions of PP-InsPs, phosphate groups immediately adjacent to the pyrophosphate moiety/moieties were not considered for derivatization. Consequently, the emerging target structures were 1PCP-InsP$_5$, derivatized at the 3- or 5-position, and 1,5(PCP)$_2$-InsP$_4$, derivatized at the 3-position (Fig. 2a). Finally, we sought to replace the primary amine used in our previously developed PCP-InsP affinity reagents[49,50] with a biotin moiety to enable resin attachment and facilitate the immobilization step[51].

Because the affinity reagents for 1PCP-InsP$_5$ and 1,5(PCP)$_2$-InsP$_4$ are asymmetrical, a synthetic strategy to obtain enantiopure material was needed. Starting from *myo*-inositol, a racemic mixture of tert-butyl-dimethylsilyl (TBS)- and para-methoxybenzyl (PMB)-protected compounds 5a and 5b could be prepared in six steps (Fig. 2b)[52,53]. The two enantiomers were separated on a gram scale using a chiral aromatic stationary phase column (Supplementary Fig. 1a). The two enantiomers were assigned through phosphitylation of the free hydroxyl group, followed by global deprotection, yielding the enantiomerically pure 1-InsP$_1$ and 3-InsP$_1$. Finally, optical rotations of these compounds were measured to complete the assignment of 5a and 5b (Supplementary Fig. 1b).

Enantiomer 5b was then further used for the synthesis of biotin-1PCP-InsP$_5$ (Fig. 2b). The free hydroxyl group at the 1-position was reacted with phosphoramidite 15, followed by oxidation to yield compound 6. Deprotection of the TBS groups and subsequent addition of one equivalent of phosphoramidite 17, followed by oxidation, yielded compound mixture 7, in which the linker was attached either at the 3- or 5-position. To increase the solubility in organic solvents after deprotection, the remaining hydroxyl group was modified with phosphoramidite 16 to provide compound mixture 8. Subsequently, the PMB groups were removed, and the resulting triol was phosphitylated and oxidized to furnish fully protected compounds 9. Final deprotection via palladium-catalyzed hydrogenolysis yielded linker-derivatized 1PCP-InsP$_5$ (10), modified at either the 3- or the 5-position. The primary amine in the linker was then coupled to NHS-biotin to provide the final product mixture of b-1PCP-InsP$_5$ (1a + 1b).

The chemical structures of compound mixture 1a + 1b were additionally investigated by 2D $^{31}$P-HMBC, as well as $^1$H-$^{13}$C-DEPT-CLIP-COZY NMR experiments (Supplementary Fig. 2). The spectra confirmed the correct attachment of the linker and PCP-groups at the desired positions and also revealed a diastereomeric mixture of 80% 1a, where the linker is attached at the 3-position, and 20% 1b, with a linker incorporated at the 5-position. Likely, the linker attachment proceeds faster at the stereochemically less hindered 3-position, due to the neighboring axial 2-OH group.

Enantiomer 5a was the central starting material for the synthesis of the 1,5(PCP)$_2$-InsP$_4$ affinity reagent (Fig. 2b). The free hydroxyl group at the 3-position was derivatized with the linker using phosphoramidite 17 and oxidized to yield intermediate 11. Following the removal of the TBS protecting groups, two PCP moieties were appended at the 1- and 5-positions utilizing phosphoramidite 15, and subsequent oxidation provided compound 12.

A 55% yield was obtained over two steps, highlighting the good compatibility of phosphoramidite 15 with challenging, sterically hindered substrates. From here on, the same synthetic sequence was applied as above, ultimately yielding enantiopure b-1,5(PCP)$_2$-InsP$_4$ (2). The structure was corroborated by 2D-NMR spectroscopy, validating the linker attachment at the 3-position and the PCP groups at the 1- and 5-positions (Supplementary Fig. 3).

To complete the series of biotinylated PCP affinity reagents, b-5PCP-InsP$_5$, and b-InsP$_6$ probes, with two alternative linker attachment sites, were also synthesized (Supplementary Fig. 4, Fig. 2a)[43,50]. All affinity reagents were quantified using $^1$H-NMR spectroscopy and an internal standard (3-(trimethylsilyl)−2,2,3,3-propanoate-d$_4$) and

prepared as 1 mM stock solutions for subsequent experiments. The described synthesis of enantiomer 5a also provided a useful starting material to access soluble 1PCP-InsP$_5$ (Supplementary Fig. 5). Combined with the previously described syntheses of 1,5(PCP)$_2$-InsP$_4$ and 5PCP-InsP$_5$, all non-hydrolyzable analogs were obtained in good quantities[54].

In summary, a synthetic strategy relying on enantiomer separation was developed to provide 1PCP-InsP$_5$ and 1,5(PCP)$_2$-InsP$_4$ affinity reagents. With this series of reagents in hand, we sought to use them for global analysis of PP-InsP-protein interactions.

## Validation and dose-dependent binding of PCP-InsP affinity reagents

We next investigated the ability of these affinity reagents to retain known protein interaction partners. The reagents were immobilized on streptavidin-coated Sepharose beads, and DIPP1, the SPX domain of XPR1 (XPR1$^{SPX}$), and the C2B domain of synaptotagmin 1 (SYT1$^{C2B}$) were subsequently incubated with the different beads. Unmodified streptavidin beads (Ctrl) were handled in parallel as control experiments. Following incubation and washing, the bound proteins were eluted with an excess of the corresponding (PCP)-InsP ligand. For all three proteins, strong retention by the four different affinity reagents, but not the unmodified beads, was observed (Fig. 3a). The isolation of target proteins by the affinity reagents could be suppressed by adding excess ligand during the binding step, consistent with competition of resin-bound (PCP)-InsP and free ligand for the same binding site. Effective elution from the resin with excess ligand was confirmed using biotin−streptavidin disrupting conditions (Supplementary Fig. 6a).

Next, we evaluated how the affinity reagents retained proteins from cell lysates. Given the high negative charge density of PP-InsPs, these molecules interact strongly with di- and trivalent metal cations[53,55,56]. Since the formation of these metal complexes likely influences the binding preferences towards different proteins, HEK293T cell lysates were prepared with 1 mM EDTA to deplete di- and trivalent cations, or with 1 mM MgCl$_2$ present (Mg$^{2+}$ ions were chosen because they are the most abundant divalent metal ions in cells)[57]. The lysates (1 mg/mL) were incubated with immobilized b-5PCP-InsP$_5$, and, following a washing step, the bound fractions were eluted with an excess of 5PCP-InsP$_5$ (Fig. 3b). The banding patterns of the eluted proteins displayed differences between the two conditions. Additionally, western blot analysis of two known binding proteins, DIPP1 and inositol polyphosphate 5-phosphatase K (INPP5K)[50,58], showed enrichment under both conditions, albeit with varying retention. The qualitative analysis of the elution profiles illustrates that the strength of PP-InsP-protein interactions is notably influenced by the presence of Mg$^{2+}$ ions. The efficiency of the binding and elution steps for these lysate experiments was further validated using an excess of competing ligand during binding, and biotin−streptavidin disrupting conditions following elution (Supplementary Fig. 6b).

Because the amount of biotinylated PCP-InsP reagents can be readily altered during immobilization, we next investigated the retention of proteins from cell lysates at different reagent concentrations. A three-fold dilution series (between 300 μM and 140 nM) of b-InsP$_6$ and b-5PCP-InsP$_5$ was prepared and immobilized, and then incubated with HEK293T cell lysate (under metal-depleted conditions). The eluates were analyzed by western blot for known binding proteins (DIPP1, SYT1, and COP9 signalosome complex subunit 5 (COPS5) for immobilized b-InsP$_6$; and DIPP1, SYT1, and ribose-phosphate pyrophosphokinase 1 (PRPS1) for immobilized b-5PCP-InsP$_5$)[50,59]. While concentration-dependent enrichment was observed for all analyzed proteins, the apparent affinities towards InsP$_6$ and 5PCP-InsP$_5$ varied (Fig.3c, Supplementary Fig. 7). For example, DIPP1 was retained more strongly by resin-bound b-InsP$_6$ compared to COPS5 and SYT1, as evidenced by the elution profile (Fig. 3c). On the other hand, for immobilized b-5PCP-InsP$_5$, SYT1 exhibited a higher affinity towards the

**Fig. 2 | Synthetic route for biotin-(PCP)-InsP compounds. a** Overview of biotin-(PCP)-InsP compounds. **b** Synthesis of b-1PCP-InsP$_5$ (**1a** + **1b**) and b-1,5(PCP)$_2$-InsP$_4$ (**2**): (i) **15**, 5-phenyl-1H-tetrazole, CH$_2$Cl$_2$ then mCPBA, 49% (ii) TBAF, THF, 75%. Afterward, **17**, 5-phenyl-1H-tetrazole, CH$_2$Cl$_2$, then mCPBA, directly used as crude (iii) **16**, tetrazole, ACN, then mCPBA, 56% over four steps, (iv) TFA in CH$_2$Cl$_2$, then

**16**, tetrazole, ACN, then mCPBA, 17% (v) Pd/C, tBuOH/H2O, 83% (vi) NHS-biotin, KH$_2$PO$_4$, H$_2$O, 81% (vii) **17**, tetrazole, ACN then mCPBA, 58% (viii) TBAF, DMF, 97%. Afterward, **15**, tetrazole, ACN, then mCPBA, 57% (ix) TFA in CH$_2$Cl$_2$, then **18**, tetrazole, ACN, then mCPBA, 71% (x) Pd/C, tBuOH/H$_2$O, 98% (xi) NHS-biotin, KH$_2$PO$_4$, H$_2$O, 58%.

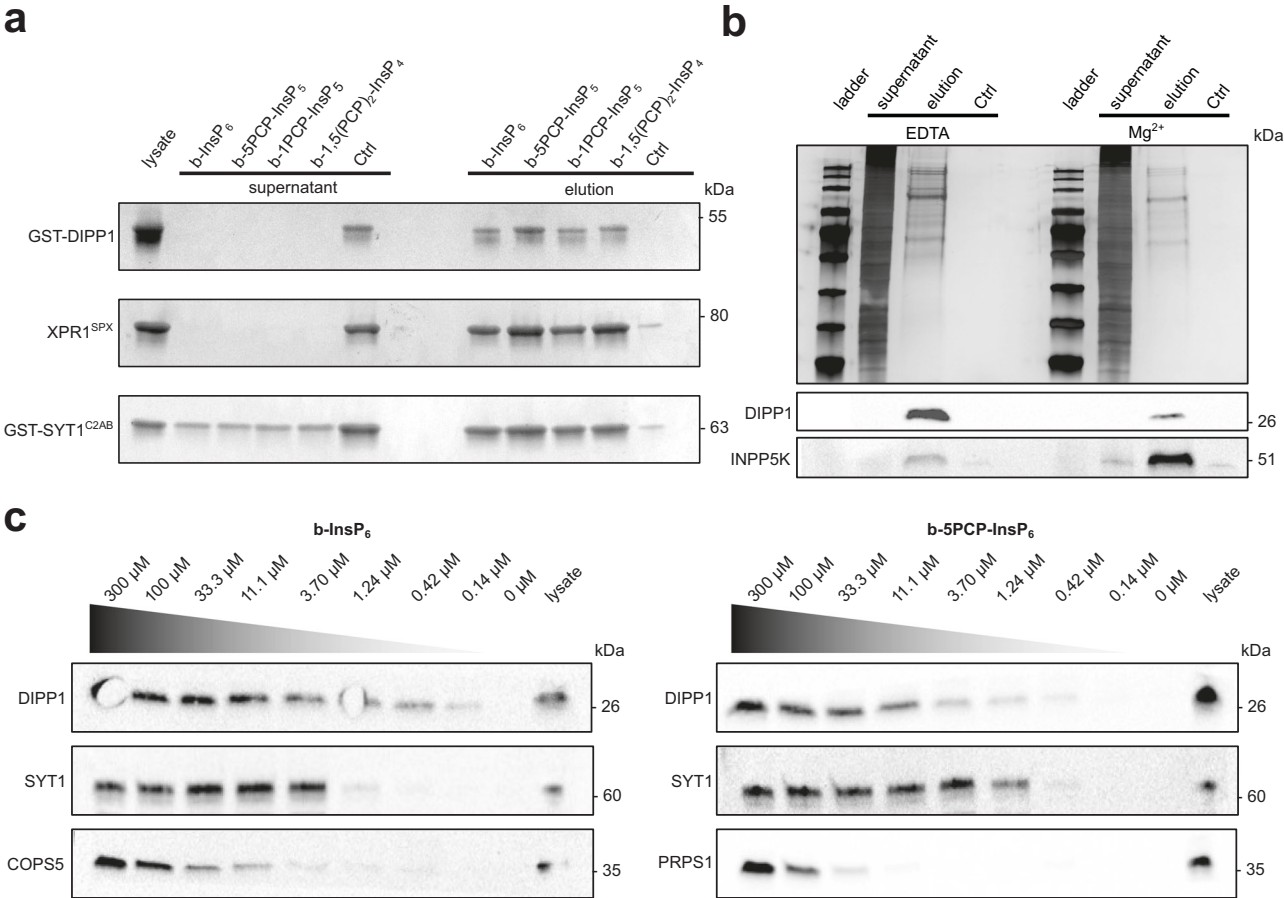

**Fig. 3 | Validation of affinity reagents and concentration-dependent affinity enrichment. a** Biochemical validation of affinity reagents b-1PCP-InsP$_5$ (**1**), b-1,5(PCP)$_2$-InsP$_4$ (**2**), b-5PCP-InsP$_5$ (**3**), and b-InsP$_6$ (**4**), (20 nmol each). Each affinity reagent was immobilzed and incubated with 5 nmol of DIPP1 (GST-DIPP1), 5 nmol of SPX domain of XPR1 (His$_6$-MBP-XPR1$^{SPX}$), or 5 nmol of the C2B domain of SYT1 (MBP-SYT1$^{C2AB}$). After 30 min incubation, the supernatant was collected, and the beads were washed three times. The bound fraction was eluted with an excess of InsP$_6$, 5PCP-InsP$_5$, 1PCP-InsP$_5$, and 1,5(PCP)$_2$-InsP$_4$ (5 mM each). Fractions were separated by SDS-PAGE and visualized by silver stain. Ctrl lanes denote streptavidin-coated Sepharose beads without immobilized reagents. Representative image from $n = 3$ independent experiments is shown. Source data are provided in the Source Data file. **b** HEK293T cells were lysed with two different buffers (common components: 25 mM HEPES, pH 7.4, 150 mM NaCl, 1% NP-40) containing either 1 mM EDTA or 1 mM MgCl$_2$. The resulting lysates were incubated with immobilized b-5PCP-InsP$_5$, and supernatants, as well as eluates, were separated by SDS-PAGE and visualized by silver stain. All fractions were also analyzed by western blot using antibodies targeting DIPP1 and INPP5K. Representative image from $n = 2$ independent experiments is shown. Source data are provided in the Source Data file. **c** Concentration-dependent affinity enrichment using immobilized b-InsP$_6$ and b-5PCP-InsP$_5$. HEK293T cell lysate (1 mg/mL) was incubated with a three-fold dilution series of affinity reagents for 1 h, under Mg$^{2+}$-depleted conditions. Bound fractions were washed and eluted with an excess of InsP$_6$ or 5PCP-InsP$_5$. Resulting fractions were analyzed by western blot using antibodies targeting DIPP1, SYT1, COPS5, and PRPS1. A lane with cell lysate was included as a positive control, whereas the 0 μM lane (no immobilized affinity reagent) serves as a negative control. Data are from a single experiment ($n = 1$). Source data are provided in the Source Data file.

ligand, compared to DIPP1 and PRPS1. These experiments demonstrated that a qualitative comparison of the binding affinities of individual proteins towards the affinity reagents is possible, using western blot analysis. We conclude that the PP-InsP affinity probes can be applied to reveal dose-dependent binding of various proteins. Analysis of the eluted proteins by mass spectrometry should, in principle, provide binding affinities across the whole proteome.

## A proteomics workflow for the determination of apparent binding constants

Inspired by recent progress in the identification of polyADPr binding proteins by Kliza et al.[45], we sought to combine the InsP$_6$/PP-InsP affinity probes with tandem mass tag (TMT) isobaric labeling to determine binding affinities on a proteome-wide scale (Fig. 4a)[44–46]. Cells were lysed using two different lysis buffers (containing either 1 mM EDTA or containing 1 mM Mg$^{2+}$), and the lysates were separated into nuclear and cytosolic fractions to provide deeper proteome coverage (Supplementary Fig. 8). Both fractions were incubated with a serial dilution (100 μM–5 nM) of the immobilized reagents b-InsP$_6$, b-5PCP-InsP$_5$, b-

1PCP-InsP$_5$, and b-1,5(PCP)$_2$-InsP$_4$ (Fig. 4a). Following quick washing steps, the retained proteins were eluted with an excess of the corresponding free ligand. The samples were digested with trypsin, followed by 11-plex TMT isobaric labeling, subsequent sample pooling, and LC-MS/MS analysis. To ensure robustness of the datasets, the experiments were performed in biological triplicates, and only proteins fully quantified in at least two replicates were further analyzed. Dose–response relationships were modeled by fitting Hill-like curves, and apparent dissociation constants (K$_D$$^{app}$) were calculated for proteins with acceptable fit quality ($R^2 \geq 0.9$). In addition, model validity was assessed by an F-test comparing the Hill-like fit to a null model, and only proteins with statistically significant fits ($p \leq 0.05$) were considered. Furthermore, a cutoff of log$_2$ fold-change $\geq 1$ between the endpoints was applied (Supplementary Data 1).

Exemplary curves are shown in Supplementary Fig. 9 and encompass an interaction of immobilized b-InsP$_6$ with C2 domain-containing protein 5 (C2CD5), in which the C2-domain likely binds to InsP$_6$. Furthermore, we observed tight interactions between resin-bound b-5PCP-InsP$_5$ and beta-arrestin-1 (ARRB1), as had been reported

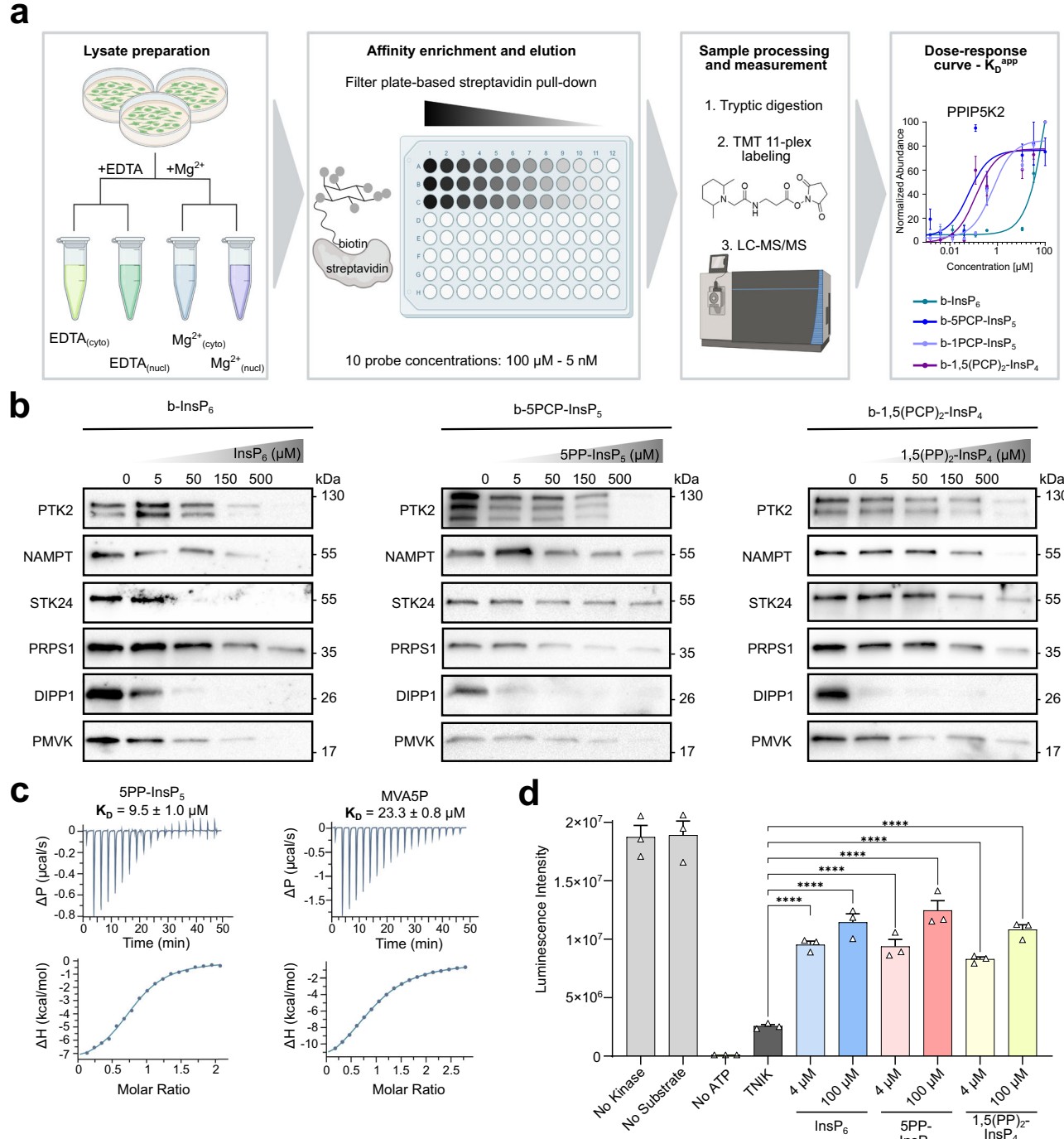

**Fig. 4 | Proteomics workflow and biochemical/functional validation of targets.**
**a** Schematic overview of the quantitative affinity enrichment workflow. Four lysate
preparations were applied to four affinity reagents (5 nM–100 μM). After elution,
samples were labeled using TMT11-plex and analyzed by LC-MS/MS. Binding curves
were used to calculate $K_D^{app}$ values. Colors indicate lysate composition/fraction:
light green, cytosolic (EDTA); dark green, nuclear (EDTA); blue, cytosolic (Mg2+);
purple, nuclear (Mg2+). Representative binding curves for each (PP)-InsP (colored)
against PPIP5K2 are shown from the Mg2+ cytosolic dataset. Data points represent
means of ≥2 biological replicates; error bars indicate SEM. Source data are pro-
vided in the Source Data file. Figure created in BioRender. Isern, J. (2026) https://
BioRender.com/5pymi74. **b** Competition-based affinity enrichment of selected
targets using b-InsP6, b-5PCP-InsP5, and b-1,5(PCP)2-InsP4. HEK293T cell lysates
(1.5 mg total protein) were preincubated with increasing concentrations of endo-
genous InsP6, 5PP-InsP5, or 1,5(PP)2-InsP4 (0–500 μM) prior to affinity purification
using immobilized biotinylated reagents on streptavidin-Sepharose beads. After

binding and washing, proteins were eluted with the respective native ligands
(10 mM), separated by SDS-PAGE, and analyzed by immunoblotting. Representa-
tive images from n = 2 independent experiments are shown. Source data are pro-
vided as a Source Data file. **c** Biophysical validation of PMVK as a ligand-binding
protein by ITC using 5PP-InsP5 and PMVK substrate MVA5P. All measurements were
performed in triplicate. **d** Functional validation of TNIK. Kinase activity was mea-
sured using the Kinase-Glo Plus assay in the presence of increasing concentrations
of InsP6 (blue), 5PP-InsP5 (red), or 1,5(PP)2-InsP4 (yellow) (0, 4, 100 μM). TNIK was
incubated with MBP (10 μM) and ATP (25 μM) for 1 h at 37 °C. Luminescence was
recorded after reagent addition. Controls lacking kinase, substrate, or ATP were
included. Data (triplicates) are shown as mean ± SEM. Statistical analysis used
ordinary one-way ANOVA with Dunnett's multiple comparisons test (each condi-
tion vs control), two-sided, with adjustment for multiple comparisons. Statistical
significance is denoted as **** ($p < 0.0001$), *** ($p < 0.001$), ** ($p < 0.01$). Source data
are provided in the Source Data file.

before[23,60], as well as strong binding between b-1,5(PCP)$_2$-InsP$_4$ and DNA-directed RNA polymerase II subunit RPB9 (POLR2I). In addition to these interactions, many known binding partners, including the (PP)-InsP metabolizing enzymes IP6K1, PPIP5K2, DIPP1, and DIPP2, were identified. As anticipated, depending on the protein interaction partner, the distinct ligands displayed a difference in $K_D^{app}$ values. For example, in the presence of Mg$^{2+}$ ions, PPIP5K2 showed tight binding to immobilized b-5PCP-InsP$_5$ and b-1,5(PCP)$_2$-InsP$_4$, while weaker binding was observed for b-1PCP-InsP$_5$ and even more so for b-InsP$_6$ (Fig. 4a).

Considering the large number of $K_D^{app}$ values that were derived from the proteomics data, we sought to validate selected interactions in binding assays, using the native InsP$_6$/PP-InsPs ligands as competitors, following previously established workflows. We chose six disease-relevant enzymes, namely focal adhesion kinase (PTK2/FAK), TRAF2- and NCK-interacting kinase (TNIK), nicotinamide phosphoribosyltransferase (NAMPT), serine/threonine-protein kinase 24 (STK24), phosphomevalonate kinase (PMVK), phosphoribosyl pyrophosphate synthase 1 (PRPS1), and DIPP1 as a control. HEK293T cell lysates were preincubated with increasing concentrations of InsP$_6$, 5PP-InsP$_5$, or 1,5(PP)$_2$-InsP$_4$, followed by incubation with the analogous immobilized probes. Upon capture and washing, bound proteins were eluted with excess ligand and analyzed by western blotting. Unfortunately, no suitable antibody targeting TNIK was found. For the remaining targets, as expected, an increase in the concentration of the competing native ligand caused a reduction of protein binding by the affinity reagents, suggesting that affinity reagents and native/unmodified InsP$_6$/PP-InsPs target the same binding sites. Consistent with the proteomics data, differences in protein binding were observed for distinct proteins and between affinity reagents. Taken together, these results validate the use of InsP$_6$/PP-InsP analogs as selective capture reagents and suggest that the observed interactomes can reflect bona fide ligand-binding events (Fig. 4b).

To further validate the binding interactions, we used ITC to determine the dissociation constants of recombinantly expressed PMVK with InsP$_6$, 5PP-InsP$_5$, and 1,5(PP)$_2$-InsP$_4$ as ligands. InsP$_6$ bound most strongly ($K_D = 4.0 \pm 0.3\,\mu M$), and affinity decreased as overall negative charge increased: 5PP-InsP$_5$, $K_D = 9.5 \pm 1.0\,\mu M$; and 1,5(PP)$_2$-InsP$_4$, $K_D = 20.6 \pm 3.4\,\mu M$ (Fig. 4c, Supplementary Fig. 10). Notably, all affinities exceeded that of PMVK's canonical substrate, 5-phosphomevalonate ($23.3 \pm 0.8\,\mu M$). We also wanted to evaluate a potential regulatory role of InsP$_6$/PP-InsPs on enzymatic activity. Consequently, we tested the activity of six recombinantly expressed proteins, namely TNIK, PMVK, STK24, PTK2, NAMPT, and PRPS1, in the presence of InsP$_6$, 5PP-InsP$_5$, and 1,5(PP)$_2$-InsP$_4$. Among the kinases tested, TNIK, a serine/threonine kinase involved in the Wnt/β-catenin signaling pathway[61], showed a concentration-dependent decrease in activity upon PP-InsP addition (Fig. 4d). In contrast, the enzymatic activities of PTK2, PMVK, NAMPT, and PRPS1 were not altered upon addition of InsP$_6$/PP-InsPs, suggesting potential alternative regulatory mechanisms that could not be adequately captured by the biochemical assays.

As a final validation step, the specificity of the InsP$_6$/PP-InsP-protein interactions was assessed across the four experimental conditions, using a comparative analysis of protein enrichment at a single probe concentration ($33.3\,\mu M$). A three-way ANOVA was applied to test the statistical significance of effects of the affinity reagent used, the cellular fraction, and the Mg$^{2+}$ availability, followed by Benjamini–Hochberg correction for multiple testing (adjusted $p < 0.05$). This analysis identified 370 proteins that were significantly enriched in a reagent-dependent manner. Distinct sets of enriched proteins were also observed under both EDTA and Mg$^{2+}$ conditions, as well as between cytosolic and nuclear fractions. Notably, retention of specific proteins between the different affinity reagents was particularly pronounced when Mg$^{2+}$ ions were present (Supplementary Fig. 11). Furthermore, relative quantitative comparisons between affinity reagents revealed reagent-specific enrichment of distinct protein sets (Supplementary Fig. 12, Supplementary Data 2). Together, this analysis highlights binding preferences for the affinity reagents and identifies Mg$^{2+}$ as an important mediator of protein–ligand interactions.

## Global analysis shows PP-InsP-specific trends

The proteomics analyses revealed a very wide distribution of $K_D^{app}$ values across the different affinity reagents, cellular fractions, and enrichment conditions. Therefore, cut-off values were defined based on the reported intracellular levels of InsP$_6$/PP-InsPs. In most human cell lines, InsP$_6$ is quite abundant with a concentration range of 10–50 μM. PP-InsPs, by contrast, are less abundant and their levels have been estimated to be 1–5 μM for 5PP-InsP$_5$, 0.1–0.5 μM for 1,5(PP)$_2$-InsP$_4$, and 0.02–0.1 μM for 1PP-InsP$_5$ (Fig. 1b)[7,48].

When the cut-off is set at $K_D^{app} < 25\,\mu M$, a total number of ca. 2700 interacting proteins across all 16 datasets were identified (Fig. 5a). This cut-off, however, is likely only relevant for InsP$_6$-binding proteins, which is why a lower cut-off $K_D^{app} < 5\,\mu M$ was implemented for the less abundant PP-InsPs. With this lower cut-off, a very different picture emerged: Within the 813 interacting proteins identified across all conditions, a large number of interactors were identified for immobilized b-5PCP-InsP$_5$ and b-1,5(PCP)$_2$-InsP$_4$ (with Mg$^{2+}$ present), and only very few proteins were enriched with the resin-bound b-1PCP-InsP$_5$ probe (Fig. 5b). In fact, the 1PCP-InsP$_5$ affinity reagent enriched only 33 proteins with a $K_D^{app} < 5\,\mu M$, from which only 2 presented a $K_D^{app} < 1\,\mu M$ (DIPP2 and its isoform DIPP2B).

The influence of coordinating divalent cations on PP-InsP-protein interactions is well apparent in the proteomics data. When the average $K_D^{app}$ values of the top 50 proteins in each dataset were examined, immobilized b-InsP$_6$ bound to proteins with higher affinities when coordinating di- and trivalent ions were depleted (Fig. 5c). In contrast, the interactions of resin-bound b-5PCP-InsP$_5$ shifted towards higher affinities with Mg$^{2+}$ ions present. For immobilized b-1,5(PCP)$_2$-InsP$_4$, the effect became even more pronounced, and all $K_D^{app}$ values were below 1 μM. These trends are exemplified by the interaction of InsP$_6$ with dedicator of cytokinesis protein 1 (DOCK1), where the interaction is weakened 20-fold by the presence of Mg$^{2+}$ ions. The opposite was observed for binding of heat- and acid-stable phosphoprotein (PDAP1) to 1,5(PCP)$_2$-InsP$_4$, where a 54-fold increase in affinity occurred upon Mg$^{2+}$ addition (Fig. 5d). Overall, the data indicates that Mg$^{2+}$-coordination of PP-InsPs appears to impose some selectivity for the recognition by target proteins (Supplementary Fig. 11). Intrigued by the high number of proteins that were enriched by 5PCP-InsP$_5$ and 1,5(PCP)$_2$-InsP$_4$ in the Mg$^{2+}$-containing nuclear fraction, we decided to further analyze these datasets.

## 5PP-InsP$_5$ and 1,5(PP)$_2$-InsP$_4$ exhibit strong Mg$^{2+}$ dependent interactions

For many protein interactions of 5PCP-InsP$_5$ and 1,5(PCP)$_2$-InsP$_4$ (in the presence of Mg$^{2+}$ ions), the $K_D^{app}$ values were below 1 μM. To rule out that the affinity enrichment is biased towards highly abundant proteins, we performed an iBAQ (intensity-based absolute quantification) analysis of a HEK293T lysate to extract the 20 proteins from the nuclear Mg$^{2+}$ fraction with the lowest $K_D^{app}$ values toward 1,5(PP)$_2$-InsP$_4$ (Fig. 6a)[27,62]. The enriched proteins were relatively evenly distributed over four orders of magnitude and included highly abundant proteins, such as pyruvate kinase (PKM) or PRPS1, and proteins of low abundance like Rho guanine nucleotide exchange factor 40 (ARHGEF40).

Given the close structural similarity between 5PCP-InsP$_5$ and 1,5(PCP)$_2$-InsP$_4$, we examined the extent of overlap between the proteins targeted by these two ligands. Among the top 200 proteins, ranked by $K_D^{app}$ values, the overlap of both datasets amounted to 24% of all proteins enriched by immobilized b-5PCP-InsP$_5$. This percentage

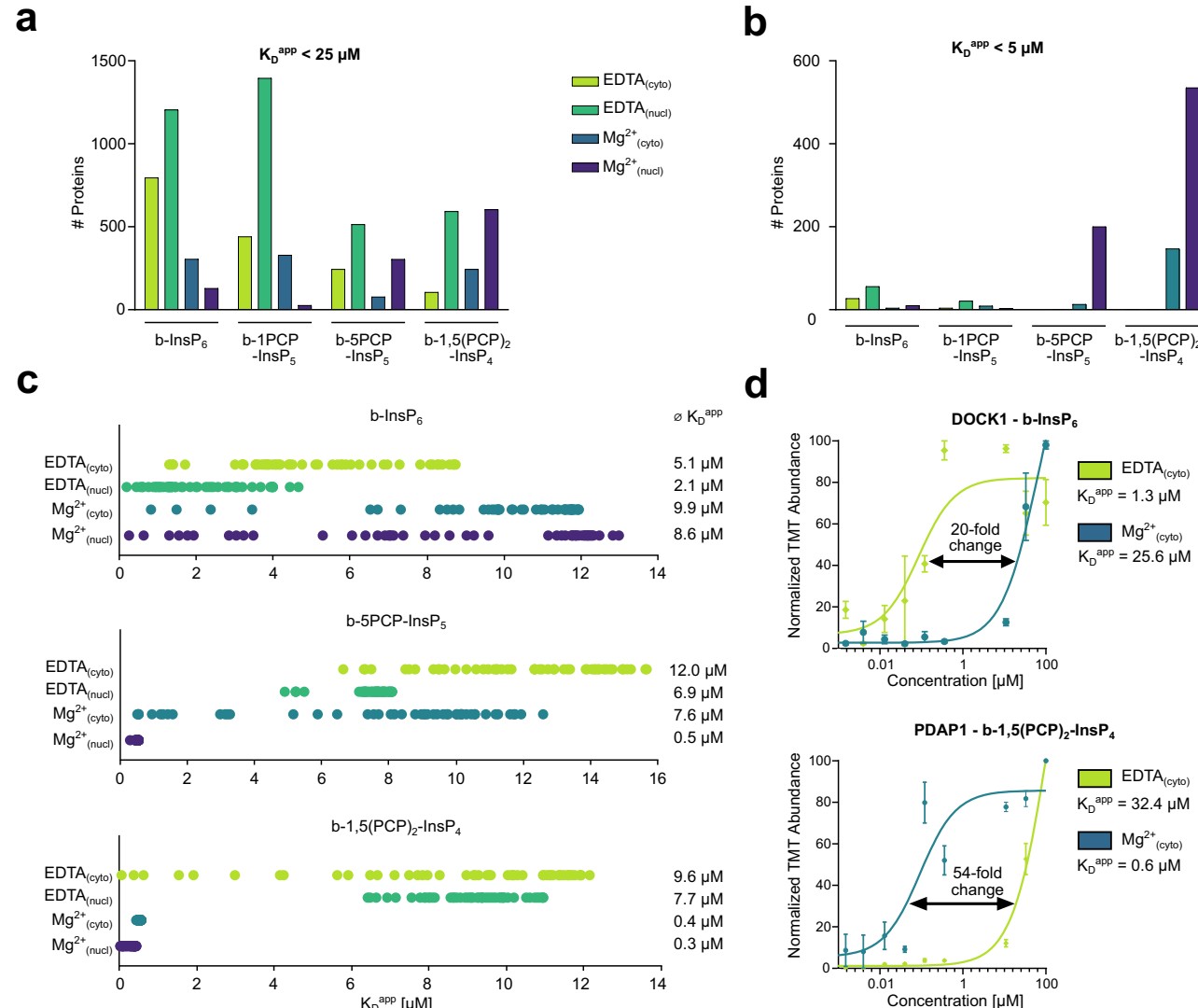

**Fig. 5 | Proteome-wide quantification of (PP)-InsP binding proteins from cell lysates. a, b** Number of all fully quantified proteins with $R^2 \geq 0.90$ for the different affinity reagents in the four lysate preparations. The cut-off for the $K_D^{app}$ values was either set to $K_D^{app} < 25\,\mu M$ (**a**) or $K_D^{app} < 5\,\mu M$ (**b**). Source data are provided in the Source Data file. **c** For the four lysate preparations, the top 50 proteins with the lowest $K_D^{app}$ values are displayed for the affinity probes b-InsP$_6$, b-5PCP-InsP$_5$, and b-1,5(PCP)$_2$-InsP$_4$. Each dot represents the $K_D^{app}$ value for a specific protein. The

average $K_D^{app}$ values (across the top 50 hits) are listed on the right side for the respective datasets. Source data are provided in the Source Data file. **d** Binding curves for the InsP$_6$ and 1,5(PCP)$_2$-InsP$_4$ interacting partners dedicator of cytokinesis protein 1 (DOCK1) and PDGFA-associated protein 1 (PDAP1). Each data point represents the mean of three biological replicates, and the error bars depict the SEM. The fold-change between the enrichment conditions (plus or minus Mg$^{2+}$) is shown in the graphs. Source data are provided in the Source Data file.

increased notably when all enriched proteins were considered without a $K_D^{app}$ cut off, 87% of the proteins enriched by immobilized b-5PCP-InsP$_5$ were also enriched by b-1,5(PCP)$_2$-InsP$_4$. These results indicate that although many proteins can interact with both PP-InsPs, the differentiation may, at least in part, be driven by their distinct binding affinities (Fig. 6b).

**Nuclear PP-InsP-binding partners are associated with RNA processing and ribosome biogenesis**

To obtain a general view of putative biological functions of 5PP-InsP$_5$ and 1,5(PP)$_2$-InsP$_4$ in the nucleus, the gene ontology (GO) terms for all 603 proteins enriched by immobilized b-5PCP-InsP$_5$ and b-1,5(PCP)$_2$-InsP$_4$ ($K_D^{app} <5\,\mu M$, Mg$^{2+}_{nucl}$, Fig. 6c) were determined (Supplementary Data 3)[63]. While we expected an enrichment of nuclear proteins, there appeared to be preferences between subnuclear compartments, including a localization to the nucleolus and the spliceosomal complex. Additionally, components of the ribosome were overrepresented among the protein interactors, which align well with the biological

processes, where many proteins involved in ribosome biogenesis, rRNA processing, and cytoplasmic translation were identified. Also, the molecular functions were consistent with these observations and included ribosome binding and rRNA binding (Fig. 6c). Overall, the data implies a function for 5PP-InsP$_5$ and 1,5(PP)$_2$-InsP$_4$ in ribosome biogenesis and transcriptional processes. Supporting these observations, a role for 5PP-InsP$_5$ in regulating the transcription of rDNA was recently reported, putatively via pyrophosphorylation of several factors that localize to the nucleolus[49,50].

Considering the GO analysis in the context of several prior studies, the nucleolus stands out as an interesting compartment where regulation by 5PP-InsP$_5$—potentially via pyrophosphorylation—can take place[27,29,64,65]. Therefore, we analyzed whether any of the recently identified pyrophosphoproteins were enriched by immobilized b-5PCP-InsP$_5$ and/or b-1,5(PCP)$_2$-InsP$_4$ in the Mg$^{2+}$ containing nuclear sample, and found that this was indeed the case for 24 out of the 59 known pyrophosphoproteins identified by Morgan et al.[27]. (Fig. 6d). Among the proteins retained by the affinity reagents were multiply

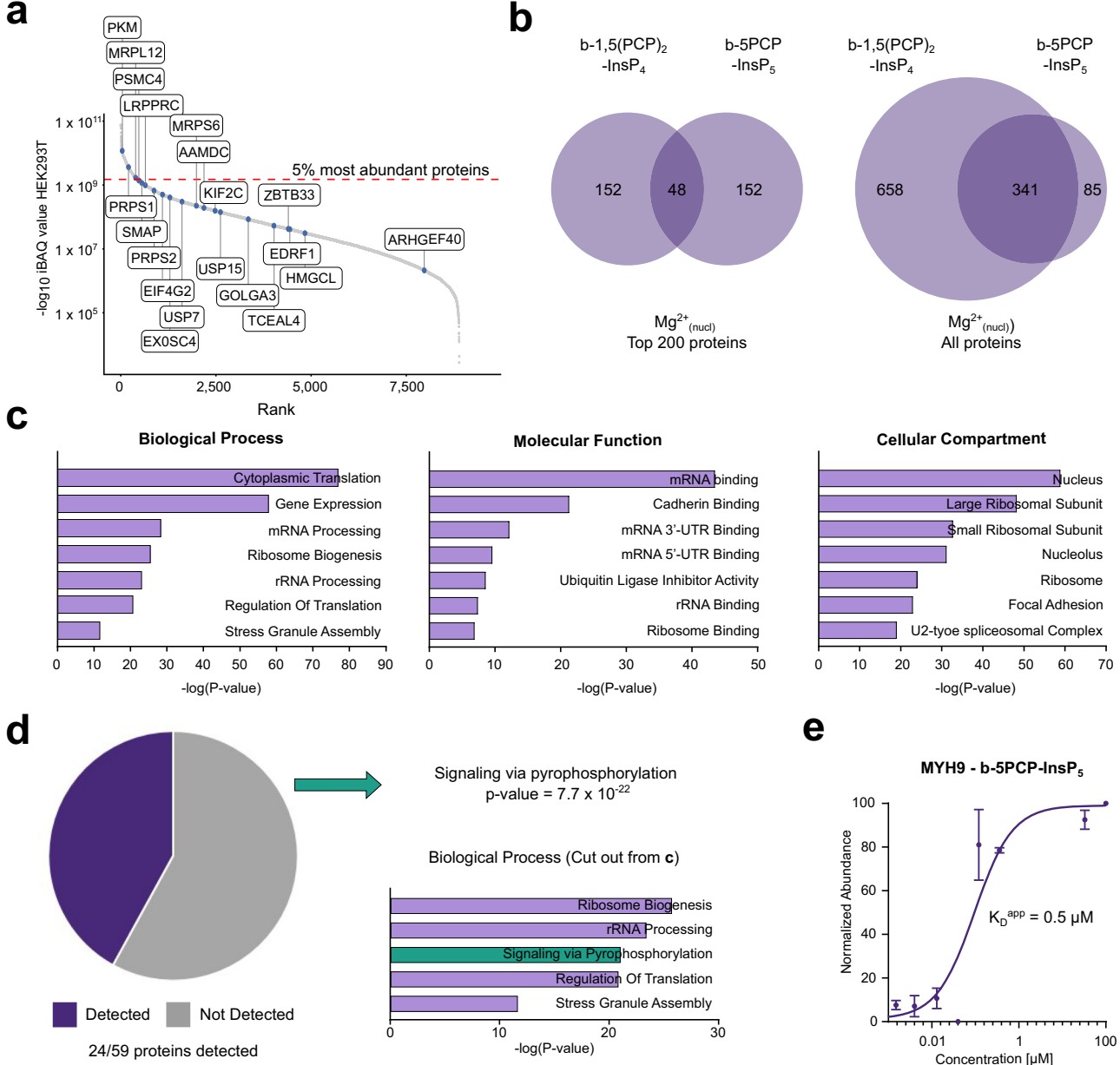

**Fig. 6 | Associated functions for nuclear proteins interacting with 5PCP-InsP$_5$ and 1,5(PCP)$_2$-InsP$_4$. a** iBAQ values of a whole HEK293T proteome were ranked and plotted against their −log$_{10}$ iBAQ values. The top 20 proteins displaying the lowest K$_D^{app}$ values that were enriched by b-1,5(PCP)$_2$-InsP$_4$ from the Mg$^{2+}$$_{nucl}$ lysate are highlighted in the graph. **b** Venn diagrams comparing the number of proteins enriched from Mg$^{2+}$$_{nucl}$ lysate between b-5PCP-InsP$_5$ and b-1,5(PCP)$_2$-InsP$_4$. Either the top 200 proteins ranked by K$_D^{app}$ values were considered (left), or all proteins for which binding curves could be determined were included (right)[82]. **c** Gene ontology (GO) analysis for the proteins enriched by b-5PCP-InsP$_5$ and b-1,5(PCP)$_2$-InsP$_4$ (K$_D^{app}$ < 5 μM, Mg$^{2+}$$_{nucl}$). The enrichment analysis was performed using Enrichr[63], applying a one-sided Fisher's exact test with Benjamini–Hochberg FDR correction for multiple testing. Source data are provided in the Source Data file.

**d** Pie chart of all proteins enriched by b-5PCP-InsP$_5$ and/or b-1,5(PCP)$_2$-InsP$_4$ in the Mg$^{2+}$$_{nucl}$ lysate with a K$_D^{app}$ < 5 μM that were also identified as pyrophosphoproteins in a previous pyrophosphoproteomics study of HEK293T cell lysates[29]. A new GO term, "signaling via pyrophosphorylation" (green), was calculated and added to the biological processes to illustrate overrepresentation. The new *p*-value was calculated using a one-sided hypergeometric over-representation test (equivalent to Fisher's exact test) to assess enrichment of the custom gene set within the query list. No adjustment for multiple comparisons was applied, as a single hypothesis was tested. Source data are provided in the Source Data file. **e** Binding curve for the 5PCP-InsP$_5$ binding protein myosin 9 (MYH9) from Mg$^{2+}$$_{nucl}$ lysate. Every data point represents the mean of three biological replicates, and the error bars depict the SEM. Source data are provided in the Source Data file.

pyrophosphorylated proteins, such as nucleolar and coiled-body phosphoprotein 1 (NOLC1) and treacle protein (TCOF1), but also proteins with only one identified pyrophosphorylation site, like myosin 9 (MYH9; Fig. 6e). To capture protein pyrophosphorylation within the GO analysis, the GO term "signaling via pyrophosphorylation" was created and assigned to the 59 known pyrophosphoproteins. When this term was included in the GO analysis above, signaling via pyrophosphorylation emerged as a highly enriched term (*p*-value of $7.8 \times 10^{-22}$). The significant overrepresentation of pyrophosphorylated proteins in the proteomics data sets consequently raised the possibility that additional pyrophosphoproteins may be found among 5PP-InsP$_5$/1,5(PP)$_2$-InsP$_4$ interactors. Thus, the affinity enrichment data were used in the following to test whether some interactors, which are not known to have a pyrophosphorylation site, are indeed pyrophosphorylated, using a targeted MS approach.

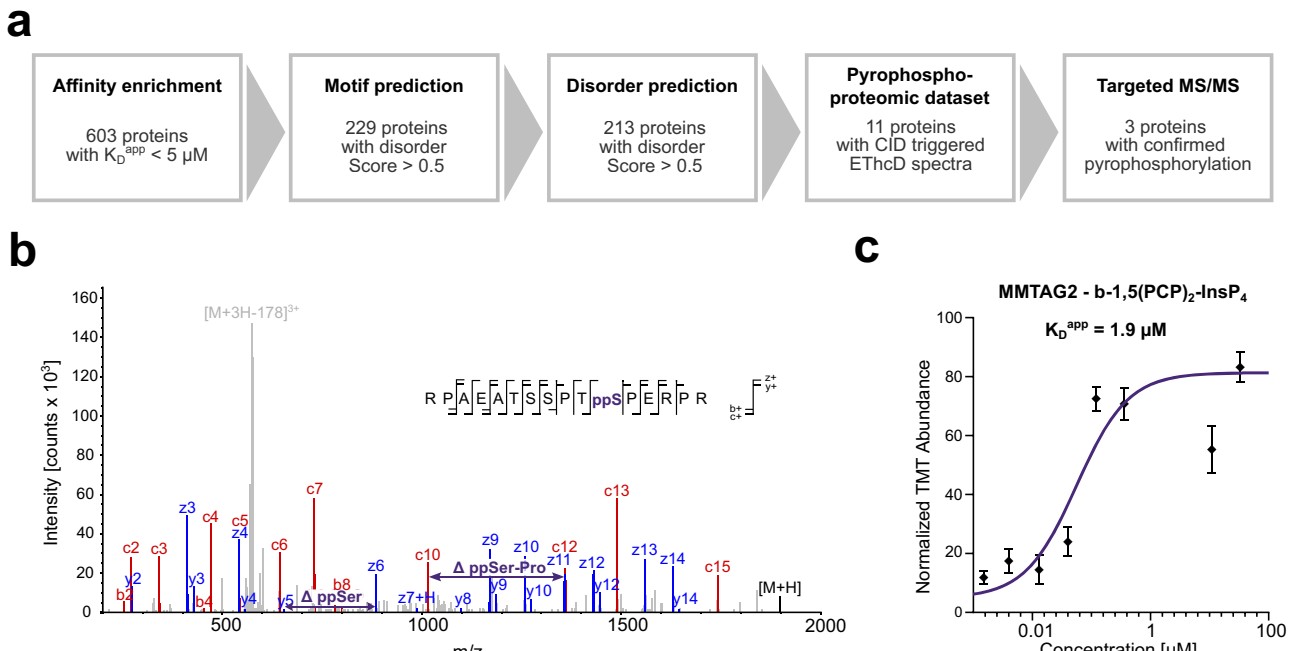

**Fig. 7 | Targeted search for pyrophosphorylated proteins. a** Workflow to predict potential pyrophosphorylated proteins from the affinity enrichment data. All 603 proteins enriched by b-5PCP-InsP$_5$ and b-1,5(PCP)$_2$-InsP$_4$ ($K_D^{app}$ < 5 µM) from Mg$^{2+}_{nucl}$ lysate were searched for acidophilic or proline-directed serine/threonine kinase site(s) and disordered regions. Subsequently, the 213 proteins were compared with a previous global MS/MS analysis, and the resulting 11 proteins were specifically targeted in an MS approach, confirming three pyrophosphoproteins. **b** EThcD spectrum of the pyrophosphorylated peptide of multiple myeloma tumor-

associated protein 2 (MMTAG2) obtained from a complex sample using a targeted MS approach. Fragment ions that are critical for the assignment of the pyrophosphorylation site at Ser220 are shown in red (b- and c-ion series) or blue (y- and z-ion series). **c** Binding curve for the enrichment of MMTAG2 with b-1,5(PCP)$_2$-InsP$_4$ from Mg$^{2+}_{nucl}$ lysate. Every data point represents the mean of three biological replicates, and the error bars depict the SEM. Source data are provided in the Source Data file.

## Enriched proteins can be linked to pyrophosphorylation–based signal transduction

Pyrophosphorylation sites are typically located within intrinsically disordered regions (IDR) and are pre-phosphorylated by acidophilic or proline-directed Ser/Thr kinases[27,28]. We therefore tested the 603 proteins that were enriched with resin-bound b-5PCP-InsP$_5$ and b-1,5(PCP)$_2$-InsP$_4$ (Mg$^{2+}_{nucl}$, $K_D^{app}$ < 5 µM) for acidophilic or proline-directed Ser/Thr kinase motifs using Scansite 4.0 (www.scansite4.mit.edu)[66], followed by an analysis of protein disorder with IUPred (www.iupred3.elte.hu/)[67]. About 35% (213 proteins) of the enriched targets fulfilled both criteria (Fig. 7a). The proteins were then compared with previous pyrophosphoproteomic analyses of HEK293T cell lysates[27], revealing that 11 proteins had tryptic peptides that displayed a characteristic neutral loss during collision-induced dissociation (CID), which subsequently triggered an electron transfer high-collision dissociation (EThcD) scan. However, these putative pyrophosphopeptides could not be confirmed in the manual assignment by Morgan et al.[27]. afterwards. Site-assignment was not possible because the EThcD spectra showed insufficient sequence coverage, with crucial fragments missing due to co-elution with corresponding bisphosphorylated peptides, or the peptides exhibited poor sequence-dependent ionization and fragmentation. We reasoned that a targeted strategy for these peptides would increase the likelihood of unambiguously detecting pyrophosphorylation sites by boosting spectral counts for each target peptide and mitigating interference from co-eluting species.

A HEK293T lysate was enriched for pyrophosphopeptides, according to Morgan et al.[27], and measured using the data-dependent neutral loss triggered EThcD method, adapted with a targeted inclusion mass list for the 11 candidate proteins. Using this approach, three proteins with four modification sites could indeed be confirmed to be

pyrophosphorylated. Exemplarily, the obtained EThcD spectrum of the multiple myeloma tumor-associated protein 2 (MMTAG2) is shown in Fig. 7b, confirming the pyrophosphorylation site on serine 220. The corresponding affinity curve of MMTAG2 binding to immobilized b-1,5(PCP)$_2$-InsP$_4$ indicated a $K_D^{app}$ value of 1.9 µM (Fig. 7c). The other confirmed pyrophosphoproteins are RNA polymerase-associated protein CTR9 homolog (CTR9; Ser 1017 and Ser 1021), and pre-mRNA-splicing factor ATP-dependent RNA helicase DHX16 (DHX16; Ser 103). The results demonstrate that potential pyrophosphoproteins can be identified using the affinity enrichment method, which can complement the mass spectrometry pipeline currently used for the detection of protein pyrophosphorylation.

## Discussion

Affinity enrichment is a well-established technique in chemical biology, frequently employed to identify binding partners of biologically relevant molecules. In many cases, the binding affinities of these interactions are of high relevance, as they determine the strength and duration of the interaction and are central to comprehending the underlying molecular mechanisms. However, binding affinities are still predominantly determined at the biochemical and biophysical level, using purified proteins. Here, we introduce the ability to determine binding affinities using a quantitative mass spectrometric approach and applied this method to the group of inositol pyrophosphate messengers.

A desymmetrization strategy was employed to generate b-1PCP-InsP$_5$ and b-1,5(PCP)$_2$-InsP$_4$ affinity reagents, immobilized via a biotin-PEG$_5$ linker. Application of these reagents to cell lysates, alongside b-InsP$_6$ and b-5PCP-InsP$_5$, demonstrated concentration-dependent affinity enrichment of known binding proteins using immunoblotting. These results motivated the application of mass

spectrometry to determine apparent binding constants on a proteome-wide scale.

The proteomics data underscore the advantages of dose–response chemoproteomics, as implemented via concentration-dependent, TMT-based affinity profiling, over traditional single-concentration assays. By fitting titration curves, $K_D^{app}$ provides an intrinsic measure of affinity that is decoupled from protein abundance and assay dose, enabling direct comparison of targets within and across studies, while potentially enhancing hit specificity. Curve fitting also improves statistical robustness (multiple data points per protein) and allows shape-based filtering that can separate specific binders from background. Unlike single-concentration experiments, the resulting affinity matrix can be used for hierarchical clustering, and structure–activity relationship analyses, improving interpretation of binding strength and selectivity, advantages previously demonstrated in related proteomics workflows[44–46,68,69]. Importantly, the apparent dissociation constants derived here reflect assay-specific conditions, including reagent concentrations, ligand derivatization, and non-equilibrium conditions, and will therefore deviate from thermodynamic $K_D$ values. Nonetheless, when determined under identical experimental conditions, $K_D^{app}$ remains a robust comparative parameter for relative binding affinity. Altogether, this strategy offers a reliable framework for annotating target engagement and thereby provides the opportunity to uncover mechanistic insights from large-scale proteomics data.

The $K_D^{app}$ proteomics analysis was carried out using a mammalian cell lysate, which had been separated into cytosolic and nuclear fractions, and which either contained $Mg^{2+}$ ions or was depleted for di- and trivalent cations. The performance of the reagents and the quantitative nature of the method were confirmed by the retention and $K_D^{app}$ determination of the $InsP_6$/PP-InsP metabolizing enzymes IP6K1, PPIP5K2, DIPP1, and DIPP2[70]. To validate some of the identified protein-ligand interactions and assess their functional relevance, competition-based affinity enrichment experiments were performed. Addition of $InsP_6$/PP-InsPs during the enrichment dose-dependently reduced protein retention by the affinity reagents, implying that the affinity reagents and native ligands bind to the same sites. Moreover, functional assays revealed that TNIK kinase activity is reduced in an $InsP_6$/PP-InsP-dependent manner, suggesting a regulatory role for these molecules. In contrast, PMVK activity was unaffected by the addition of $InsP_6$/PP-InsPs, although ITC measurements confirmed binding to $InsP_6$/PP-InsPs. These validation experiments demonstrate that individual PP-InsP-ligand interactions derived from the proteomics data should always be confirmed by biochemical and/or biophysical methods and that different modes of action may be uncovered for $InsP_6$/PP-InsPs. Full characterization of these interactions will require further functional validation in cellula, for example, by introducing point mutations into the putative $InsP_6$/PP-InsP binding pockets. To associate specific PP-InsPs with these regulatory events, the mutagenesis studies could be coupled to selective perturbations in PP-InsP biosynthesis. In this fashion, specific PP-InsPs can be placed into distinct cellular pathways. It is envisioned that the data provided here will spark the experimental validation for these complex functional associations.

As anticipated, a wide range of apparent binding constants was observed in our proteomics experiments, ranging from $K_D^{app}$ values between 11 nM and 25 μM. To aid the subsequent analysis, we defined cut-off values based on average cellular concentrations of $InsP_6$/PP-InsPs. It is important to note that these cut-off values are somewhat arbitrary and cannot accurately reflect the local concentration of these messengers. Even though many observed binding interactions lie above the defined thresholds, these interactions may still be biologically relevant, given the appropriate cellular surroundings. A method that reliably measures the concentration of the different PP-InsPs with subcellular resolution—akin to the reporters used for phosphatidyl

inositols—would be of great use. Such a reporter could also provide insight into another unexplained phenomenon: many interactions of high affinity were observed for $1,5(PP)_2$-$InsP_4$ with nucleolar proteins, yet it is not known if this messenger actually localizes to this compartment in intact cells.

Not only is the arrangement of the pyrophosphate groups around the *myo*-inositol scaffold important for recognition, but also the speciation (i.e., protonation or metal-coordination) of the different PP-InsPs. The highly phosphorylated inositols have a special relationship with $Mg^{2+}$ ions, the most abundant divalent metal ions in cells[58]. Coordination of $Mg^{2+}$ ions strongly influences the solubility of $InsP_6$/PP-InsPs, their charge, and their hydration shells, which all, in turn, can have an effect on the binding strength towards different target proteins[53,55,71]. Depending on the binding sites that are targeted by the PP-InsPs, it may be necessary to remove the $Mg^{2+}$ ions for protein binding—a process that will likely be energetically costly. In the structurally characterized examples of PP-InsP-protein complexes, $Mg^{2+}$ ions are rarely found. Instead, a large number of positively charged side chains are involved in neutralizing the charge of the highly negatively charged ligand. It would therefore be interesting to investigate whether cells can employ regulatory mechanisms to control local $Mg^{2+}$ ion availability, thereby strengthening (or weakening) a specific set of PP-InsP protein interactions.

Notably, many PP-InsP-protein interactions were strengthened by the presence of $Mg^{2+}$ ions, suggesting that the metal ions may play a role in stabilizing either the protein, the PP-InsP ligand, or the protein-ligand interface. A possible scenario is that certain proteins may actually prefer binding to PP-InsPs in a $Mg^{2+}$-coordinated, flipped conformation, in which five substituents of the *myo*-inositol ring are placed in axial positions. It was previously reported that app. 30% of $1,5(PP)_2$-$InsP_4$ adopts such a flipped conformation in the presence of $Mg^{2+}$ ions[55]. However, to date, no characterized example of a flipped PP-InsP-protein complex exists. With the recent advancement of high-resolution structural methods, such as cryo-electron microscopy and NMR spectroscopy, such characterization now seems feasible.

Another scenario is that $Mg^{2+}$ ions help to bring together the negatively charged PP-InsPs with protein sequences that are targets of pyrophosphorylation, as pyrophosphorylation sites are commonly found in serine-rich polyacidic stretches[27,28]. $Mg^{2+}$ ions have been reported to be essential for these unusual phosphoryl transfer reactions and likely act as a molecular glue for the two negatively charged reaction partners. Consistent with this, numerous pyrophosphorylation targets were enriched with b-5PCP-$InsP_5$ and b-$1,5(PCP)_2$-$InsP_4$ in the presence of $Mg^{2+}$, which implies a specific interaction between the PP-InsP-magnesium complex and negatively charged pyrophosphorylation sequences. A similar observation was made previously for 5PP-$InsP_5$ interacting proteins from *S. cerevisiae*, where several known pyrophosphorylation targets were retained upon the addition of $Mg^{2+}$ ions[49]. Although the involvement of $1,5(PP)_2$-$InsP_4$ in pyrophosphorylation chemistry remains speculative, it appears reasonable that this molecule is also capable of transferring its β-phosphoryl group(s). Further research is required to elucidate the contribution of distinct PP-InsP messengers to protein pyrophosphorylation. With the development of new chemical methods to obtain different PP-InsP isomers, and the recent implementation of a mass spectrometry method to detect pyrophosphorylation sites, such studies should be conducted in the future[17,27].

Given the significant enrichment of pyrophosphoproteins in our data sets, we investigated the possibility that additional targets of pyrophosphorylation were retained by the affinity reagents. Indeed, by using a targeted mass spectrometry approach for 11 putative pyrophosphorylation sites, we could confirm four previously uncharacterized sites on three proteins. Because the detection of pyrophosphorylation sites on peptides remains a challenge (due to low abundance, poor ionization, and conflicting phosphorylation

patterns), it is necessary to enrich pyrophosphorylated peptides in the currently implemented mass spectrometry workflow. This enrichment mainly relies on sequential elution from immobilized metal affinity chromatography (SIMAC), followed by fractionation[27,31]. The affinity reagents presented here could offer an alternative approach to enrich pyrophosphorylation targets at the protein level, especially when applied to nuclear or nucleolar lysates.

One limitation of the current method is the capture of protein complexes, such as ribonucleoprotein complexes. Within these assembled structures, not every protein interacts with the InsP$_6$ or PP-InsP ligand. These complexes will bias the gene ontology analysis since all identified protein components are included in the data query input. The enrichment of indirect binding partners could be reduced by forming a covalent bond between the ligand and its direct binding partner, allowing for much more stringent washes. Such a covalent capture could, for example, be achieved by incorporating photo-crosslinkers in the future.

In summary, the affinity reagents and the method described here represent a versatile toolkit for the exploration of inositol pyrophosphate signaling. The proteomics data sets and the affinity reagents provide valuable community resources. These resources should now be leveraged to further discover and characterize PP-InsP-interacting proteins, to interrogate signaling pathways across different cell types and organisms. Application of affinity reagents to specific subcellular compartments, such as the nucleolus, where numerous pyrophosphoproteins localize, may uncover uncharacterized layers of spatial regulation and mechanistic insight into this unusual phosphorylation mode. Given the pleiotropic roles and evolutionary conservation of PP-InsPs, the continued use of these affinity tools can reveal previously unrecognized dimensions of their signaling networks in eukaryotic biology.

## Methods

### Chemical synthesis
List of reagents and resources used, as well as experimental details for chemical synthesis and characterization data of the compounds, can be found in the Supplementary Information (Supplementary Table 1).

### Cell culture
HEK293T cells (female human origin) were cultured in Dulbecco´s Modified Eagles´ Medium (DMEM), with 10% FBS, Penicillin-Streptomycin (100 U/mL), and Glutamine (2 mM) in a 5% humidified $CO_2$ incubator at 37 °C. The cells were received from the American Type Culture Collection (ATCC) and tested for mycoplasma before use.

### Cell lysate preparation
Whole-cell extracts for validation experiments were prepared as described previously with certain modifications[50,72]. Briefly, HEK293T cells were grown to 90% confluency in 15 cm dishes. Cells were washed twice with ice-cold 0.9 mM NaCl in $H_2O$ (10 mL) and lysed with either Pierce™ IP Lysis buffer (TRIS EDTA conditions), buffer 2 (150 mM NaCl, 25 mM HEPES pH 7.4 at 4 °C, 1 mM EDTA, 1% NP-40, HEPES EDTA conditions) or buffer 3 (150 mM NaCl, 25 mM HEPES pH 7.4 at 4 °C, 1 mM MgCl$_2$, 1% NP-40, HEPES Mg$^{2+}$ conditions). 2.7 mL lysis

buffer, supplemented with phosphatase and protease inhibitors (Roche PhosStop™ and cOmplete™ EDTA-free protease inhibitor cocktail), was added, scraped off, added to protein low-binding microcentrifuge tubes, and incubated on ice for 10 min. The lysate was centrifuged at 4 °C for 15 min at 21.000 g. The supernatants were combined, and protein concentration was measured by the Pierce™ BCA Protein Assay kit.

Nuclear and cytosolic cell extracts for affinity enrichment experiments were prepared the following: HEK293T cells were grown to 90% confluency in 15 cm cell culture dishes. Cells were washed twice with ice-cold 0.9 mM NaCl in $H_2O$ (10 mL), carefully scraped off, added to protein low-binding microcentrifuge tubes, and ice-cold PBS (1 mL) was added. The cells were pelleted for 2 min, 250 g at 4 °C, and the supernatant was removed. The pellet was washed twice with ice-cold PBS (1 mL), the supernatant was removed, and the pellet was kept on ice. Afterward, the cells were either lysed by EDTA-containing buffer (150 mM NaCl, 25 mM HEPES pH 7.4 at 4 °C, 1 mM EDTA, 0.1% Triton™ X-100, Roche PhosStop™, and cOmplete™ EDTA-free protease inhibitor cocktail) or Mg$^{2+}$-containing buffer (150 mM NaCl, 25 mM HEPES pH 7.4 at 4 °C, 1 mM MgCl$_2$, 0.1% Triton™ X-100, Roche PhosStop™ and cOmpleteTM EDTA-free protease inhibitor cocktail). 1 mL lysis buffer was added, and the cells were incubated on a rotation wheel for 30 min. The cells were centrifuged for 2 min, 250 g at 4 °C. The supernatant, containing the cytosolic proteins, was transferred into Falcon tubes, and the lysis was repeated with 1 mL lysis buffer, incubation for 30 min and centrifugation. Subsequently, the supernatant was recovered and added to the same Falcon tube. After the lysis of the cytosolic fraction, the cell pellet was dissolved in 1 mL respective lysis buffer followed by sonication at 4 °C (on ice, 50% output, 0.5 cycle rate, 5 × 30 s, 30 s rest between pulses). The lysate was centrifuged (30 min, 21.000 g, 4 °C) and the supernatant was transferred into Falcon tubes. The fractions were aliquoted, snap-frozen, and stored at −70 °C.

The concentration of both lysate fractions was determined by the Pierce™ BCA Protein Assay kit.

### Affinity enrichment for proteomics analysis and immunoblotting
Hydrophilic PVDF Multiscreen® 96-well plates were mounted on a PlatePrep 96-well vacuum manifold. Wells were pre-washed with 70% ethanol (50 μL), followed by two washes with ice-cold PBS (200 μL each). For affinity purifications, individual biotinylated probes (b-InsP$_6$, b-1PCP-InsP$_5$, b-5PCP-InsP$_5$, and b-1,5(PCP)$_2$-InsP$_4$) were immobilized on streptavidin Sepharose beads. Streptavidin beads (80 μL per well) were added to the plate and washed three times with ice-cold PBS (200 μL). After removing residual liquid, the plate was placed on a tabletop shaker at 4 °C. Then, 30 nmol of each affinity reagent (30 μL of a 1 mM stock) were diluted in PBS to a final volume of 300 μL. For SDS−PAGE and immunoblotting experiments, reagents were diluted to 150 μL. A three-fold serial dilution was prepared as indicated in the Table 1, and all experiments were performed in triplicate. Aliquots (150 μL) of each affinity reagent dilution were added to the streptavidin bead-containing wells. Control wells received 150 μL PBS only. The plate was incubated at 300 rpm for 30 min at 4 °C, followed by washing under vacuum with ice-cold PBS (2 × 200 μL). Wells were then washed once with either EDTA-containing buffer (150 mM NaCl, 25 mM HEPES

**Table 1 | Affinity reagents serial dilution scheme**

| | 1 | 2 | 3 | 4 | 5 | 6 | 7 | 8 | 9 | 10 | 11 |
|---|---|---|---|---|---|---|---|---|---|---|---|
| μL affinity reagent | - | 100 from 3 | 100 from 4 | 100 from 5 | 100 from 6 | 100 from 7 | 100 from 8 | 100 from 9 | 100 from 10 | 100 from 11 | 30 from 1 mM stock |
| μL PBS | 150 | 200 | 200 | 200 | 200 | 200 | 200 | 200 | 200 | 200 | 270 |
| Final concentration (μM) | 0 | 0.005 | 0.015 | 0.0045 | 0.137 | 0.41 | 1.23 | 3.7 | 11.1 | 33.3 | 100 |

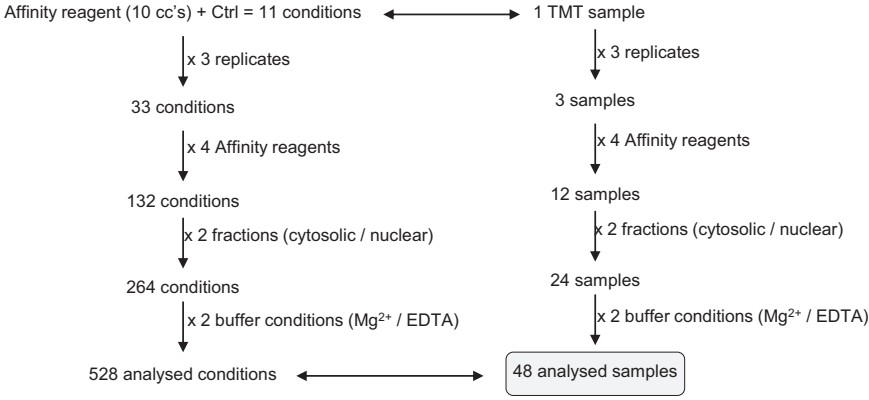

**Fig. 8** | Summary of the mass spectrometry samples and conditions analyzed.

pH 7.4, 1 mM EDTA, 0.1% Triton™ X-100) or Mg$^{2+}$-containing buffer (150 mM NaCl, 25 mM HEPES pH 7.4, 1 mM MgCl$_2$, 0.1% Triton™ X-100), both at 4 °C.

After immobilization, cell lysate (150 μL, 0.7 mg/mL) was added to each well and incubated at 300 rpm for 60 min at 4 °C. Unbound lysate was removed by vacuum filtration, and wells were washed six times with either EDTA-containing washing buffer (25 mM HEPES, pH 7.4, 1 mM EDTA) or Mg$^{2+}$-containing washing buffer (25 mM HEPES, pH 7.4, 1 mM MgCl$_2$), both at 4 °C. For competition experiments, the filter plate was placed onto a conical 96-well collection plate, and competing (PCP)-InsPs were added (100 μL, 5 mM in 25 mM HEPES pH 7.4 containing either 1 mM EDTA or 1 mM MgCl$_2$). Control experiments received a mixture of all competing InsPs (total volume 50 μL). Plates were incubated at 300 rpm for 15 min, followed by centrifugation at 1000 × $g$ for 2 min. The competition step was repeated once. Affinity-enriched eluates were subsequently processed for immunoblotting or proteomic sample preparation.

### In solution digestion and TMT labeling

Eluted proteins (100 μL per sample) were denatured by the addition of MeOH-containing buffer (100 μL, 40% MeOH, 160 mM TEAB, pH 8.5) and mixed gently. The samples were reduced by the addition of TCEP (5 mM final concentration in TEAB pH 8.5) and alkylated with CAA (40 mM final concentration in TEAB pH 8.5, prepared freshly) and incubated in the dark at rt for 1 h at 1000 rpm. LysC (1:200 in 50 mM TEAB, pH 8.5) was added and incubated for 2 h at 37 °C, 1000 rpm. Trypsin (1:100 in 50 mM TEAB, pH 8.5) was added, and the mixture was incubated at 37 °C for 16 h at 1000 rpm. Afterward, the solvents were removed by vacuum centrifugation for 7 h at 37 °C.

The dried samples were resolubilized in 20 μL TEAB pH 8.5 and gently pipet mixed. TMT11-plex (0.8 mg each) was dissolved in 42 μL water-free ACN and vortexed. To each sample, 3.3 μL of the corresponding TMT reagent was added, and the plate was incubated for 1 h at RT in the dark, 1,000 rpm on a tabletop shaker. The reaction was quenched by adding TRIS-HCl (10 μL, 1 M, pH 8.0) and incubated for 30 min at 1000 rpm at rt. Afterward, the fractions were pooled, the wells were washed again with 20 μL ACN and the solvents were removed by vacuum centrifugation for 2 h at 37 °C. Samples were dissolved in Milli Q water, acidified with 10% TFA, and desalted using StageTipping[73]. A description of the analyzed mass spectrometry samples and conditions can be found in Fig. 8.

### Immunoblot analysis

30 μL of affinity-enriched lysate were mixed with 10 μL 4x laemmli sample buffer and boiled at 95 °C for 10 min. The reagent was run on a 4−20% SDS-gel and transferred to a 0.45 μM pore size PVDF membrane (Trans turbo blot, mixed MW, 7 min). The membrane was blocked for 1 h with 5% non-fat dried milk powder in TBS-T buffer at rt. The primary

antibody in TBS-T buffer was incubated for 1.5 h and washed five times for five minutes with TBS-T buffer. Subsequently, the secondary HRP-conjugated antibody in TBS-T was incubated for 1.5 h and washed five times for five minutes with TBS-T buffer. The western blot was visualized using SuperSignal™ West Femto (Thermo Fisher Scientific) and analyzed by ImageLab 6.1.0 (BioRad).

### Recombinant protein expression

Expression was performed as previously described for GST-hDipp1[49], MBP-XPR1$^{SPX}$[37], and SYT1$^{C2B}$[74].

### Affinity enrichment with recombinant proteins

Multiscreen® 96-well Plate, hydrophilic PVDF membrane was placed on the PlatePrep 96-well vacuum distributor, and wells were washed with 70% EtOH (50 μL) followed by two washes with PBS (200 μL, 4 °C). Streptavidin sepharose was added (80 μL) and washed with ice-cold PBS (three times 200 μL). The bottom was dried with a clean wipe, and the plate was positioned on a tabletop shaker at 4 °C. 20 μL of a 1 mM b-(PCP)-InsP solution was dissolved in 150 μL PBS and added to the plate. For control experiments, 150 μL PBS was added, and the plate was incubated at 300 rpm for 30 min. The plate was placed on the vacuum and washed with ice-cold PBS (two times 200 μL) followed by one wash with ethylenediamine tetraacetic acid (EDTA) containing buffer (200 μL, 150 mM NaCl, 25 mM HEPES, pH 7.4 at 4 °C, 1 mM EDTA, 0.1% Triton™ X-100). The bottom of the plate was dried, and it was placed on the tabletop shaker at 4 °C. Recombinant protein (150 μL, 5 nmol) was added and incubated at 300 rpm for 60 min. The plate was placed on a 96-well plate and centrifuged (4 °C, 1000 g, 2 min) to collect the unbound proteins in the supernatant. The wells were quickly washed six times with the incubation buffer. The filter plate was placed on a conical 96-well plate, and competing (PCP)-InsPs were added (50 μL, 5 mM (PCP)-InsP in 25 mM HEPES, pH 7.4 at 4 °C, 1 mM EDTA). For control experiments, all competing InsPs used were added (50 μL in total). The filter plate was incubated at 300 rpm for 15 min followed by centrifugation at 1000 g for 2 min and the competition was repeated once more. The resulting affinity-enriched lysates were used for SDS-PAGE analysis. 30 μL of affinity-enriched lysate was mixed with 10 μL 4x laemmli sample buffer and boiled at 95 °C for 10 min. The reagent was run on a 4−20% SDS-gel and visualized by InstantBlue® Coomassie Protein Stain (abcam).

### Pyrophosphoproteomics sample preparation

The pyrophosphoproteomics sample preparation workflow was performed as described by Morgan et al.[27].

### Liquid chromatography and mass spectrometry

Desalted peptides were resuspended in 1% ACN with 0.05% TFA, and 1 μg was injected into a Thermo Scientific™ Dionex™ UltiMate™

3000 system connected to a PepMap C-18 trap-column (0.075 mm × 50 mm, 3 μm particle size, 100 Å pore size, Thermo Fisher Scientific) followed by an in-house packed C18 column for reverse phase separation (Poroshell 120 EC-C18, 2.7 μm, Agilent Technologies). With a flow rate of 300 nL/min, peptides were separated using a 117 min gradient with increasing ACN concentration and analyzed on an Orbitrap Fusion Lumos mass spectrometer with FAIMS Pro™ device (Thermo Fisher Scientific) and Instrument Control Software version 4.0. MS1 and MS2 scans were acquired in the Orbitrap with a mass resolution of 120,000 and 50,000 respectively. MS1 parameters were as following: scan range m/z 400–1600, standard AGC target, 246 ms maximum injection time. MS2 parameters were as following: scan range first m/z 110, AGC target 1.25e5, 86 ms maximum injection time, isolation window 0.7 m/z, NCE 38%. Previously isolated precursors were excluded from fragmentation for 60 s. Only precursors with charges +2 – +6 were subjected to MS2. Data were acquired using 2 s per column volume (CV) with an internal stepping of CVs from −50 to −65 and −85.

The LC-MS measurement for the pyrophosphoproteomics samples was performed in a similar manner as described by Morgan et al., with minor changes using a targeted data-dependent neutral loss triggered EThcD approach[27]. The inclusion list contained masses of expected pyrophosphorylated peptides. Fractionated pyrophosphoproteomics samples were resuspended with 50 mM medronic acid in 3% ACN and injected in a Thermo Scientific™ Dionex™ UltiMate™ 3000 system connected to a PepMap C-18 trap-column (0.075 mm × 50 mm, 3 μm particle size, 100 Å pore size, Thermo-Fisher Scientific) followed by an in-house packed C18 column for reverse phase separation (Poroshell 120 EC-C18, 2.7 μm, Agilent Technologies). With a flow rate of 250 nL/min, peptides were separated using a 117 min gradient with increasing ACN concentration and analyzed on an Orbitrap Fusion mass spectrometer device (Thermo Fisher Scientific) and Instrument Control Software version 4.0. MS1 scans were acquired in the Orbitrap with a mass resolution of 120,000, the MS1 parameters were as following: scan range m/z 380–1400, standard AGC target, 50 ms maximum injection time. Precursor ions were selected using targeted mass inclusion lists with an unscheduled time mode. Precursor ions with charge states 2–4 were isolated with an isolation window of 1.6 m/z and priority to the higher charge state. MS2 CID scans were acquired in the Orbitrap with a mass resolution of 15,000, the MS2 CID parameters were as following: AGC target 2.5e4, 100 ms maximum injection time, NCE 25%. If neutral losses of 177.9432 Da were measured within the Top 10 most intense ions in the CID scan, an additional spectrum of the same precursor ion was acquired using EThcD. MS2 EThcD scans were acquired in the Orbitrap with a mass resolution of 120,000, the MS2 EThcD scan parameters were as following: AGC target 1e5, 2000 ms maximum injection time, normalized supplemental activation energy 30%.

### Data analysis pyrophosphoproteomics data
Raw files were analyzed using Proteome Discoverer (ThermoFisher Scientific) version 3.0 as described by Morgan et al.[27].

### Chemoproteomic data analysis: identification and quantification of InsP$_6$/PP-InsPs binders
Proteins have been identified and quantified from RAW files using SEQUEST and MSAmanda 2.0[75] in ProteomeDiscoverer v3.0 using the following search parameters: spectrum recalibration, MS1 mass tolerance, 10 ppm; MS2 mass tolerance, 20 ppm; maximum number of missed cleavages, 2; minimum peptide length, 6; peptide-mass, 350–8000 Da. Carbamidomethylation (+57.021 Da) on cysteines was used as a static modification. Oxidation of methionines (+15.995 Da) and TMT6plex on lysines and peptide N-termini (+229.163 Da) were set as variable modifications. Data was searched against the human proteome (retrieved from Uniprot, one protein sequence per gene). The FDR has been set to 1% on the protein level using target decoy FDR nodes. TMT quantification was achieved using the reporter ions quantifier with default settings, with application of quantification value correction.

### Affinity determination using TMT abundance
Affinity determination was performed as previously described[44–46]. In brief, protein tables for each replicate were exported from Proteome Discoverer and processed in R. Biological replicates were normalized individually to their maximum values, and quantification values were merged across the three replicates. For each protein, only cases where at least two replicates were fully quantified were retained. Averaged protein abundances and corresponding standard deviations were then calculated. Dose–response relationships were modeled by fitting Hill-like curves using the drc R package[76], and apparent dissociation constants ($K_D^{app}$) were derived from the fits. Fit quality was assessed by multiple criteria: the Pearson correlation coefficient (moderate > 0.90, good > 0.95, perfect > 0.99), the coefficient of determination ($R^2 \geq 0.9$), and statistical significance from an F-test comparing the Hill-like model to a null model ($p \leq 0.05$). To ensure robustness, a cutoff of $\log_2$ fold-change $\geq 1$ between the 100 μM treatment and control was applied to define proteins exhibiting meaningful enrichment.

### GO term enrichment analysis of PP-InsP binders
GO enrichment analysis of selected proteins was carried out using Enrichr with default parameters: one-sided Fisher's exact test with Benjamini–Hochberg FDR correction for multiple testing[63].

### Tandem competitive assays coupled with affinity enrichment and western blotting
HEK293T cell lysates (1.5 mg total protein) were incubated with increasing concentrations of endogenous InsP$_6$, 5PP-InsP$_5$, or 1,5(PP)$_2$-InsP$_4$ (0, 5, 50, 150, 500 μM; final volume 600 μL) for 1 h at 4 °C with end-over-end rotation. In parallel, the corresponding biotinylated affinity reagents (30 μL, 1 mM) were diluted in 570 μL buffer and immobilized on streptavidin-Sepharose beads (100 μL per sample), prewashed three times with ice-cold PBS[77]. After 1 h rotation at 4 °C, the beads were washed three times with PBS and once with binding buffer before incubation with the pretreated cell lysates for 1 h at 4 °C under rotation. The beads were then washed three times with binding buffer, and bound proteins were competitively eluted with the corresponding native ligand (10 mM; 2 × 75 μL) for 30 min at 4 °C with shaking (950 rpm). Finally, eluates were separated by SDS-PAGE and analysed by immunoblotting using SuperSignal™ West Femto Maximum Sensitivity Substrate, as previously described with the corresponding antibodies.

### TNIK-inhibitor kinase assay
Kinase assays were conducted using a Kinase-Glo Plus luminescence assay kit (Promega). Recombinant TNIK (1-367) (2 ng/mL) was added to the reaction buffer (40 mM Tris-HCl, pH 7.5, 4 mM MgCl$_2$, 1 mM MnCl$_2$, 0.5 mM DTT, 0.1 mg/mL BSA) and preincubated for 30 min in the presence of increasing concentrations of InsP$_6$, 5PP-InsP$_5$ and 1,5(PP)$_2$-InsP$_4$ (0, 4, 100 mM). Subsequently, bovine myelin basic protein (MBP) (10 mM) and ATP (25 mM) were added to a final volume of 25 μL, and the reaction mixtures were incubated for 1 h at 37 °C. Finally, the reaction was stopped by the addition of Promega Kinase-Glo Plus reagent (25 μL) and luminescence read out by luciferase activity with a with a Tecan Infinite M Plex reader using 1000-ms exposure time after 5 min of equilibration. As control, no kinase, no MBP and no ATP conditions were also assayed. All experiments were conducted in triplicate.

## Probe vs probe relative quantification

The enrichment specificity of affinity reagents b-InsP$_6$, b-1PCP-InsP$_5$, b-5PCP-InsP$_5$ and b-1,5(PCP)$_2$-InsP$_4$ at 33.3 μM in HEK293T was assessed across all conditions (cytosolic/nuclear fractions under Mg$^{2+}$/EDTA). Baits were quantitatively compared against each other using the data obtained from the proteomic assays previously described. All experiments were performed in biological replicates. Protein tables obtained as previously described were processed in R (v4.5.0). Proteins were required to be quantified in at least 2 of 3 replicates in either condition to enter downstream analysis. Intensities of zero were set to missing (NA), and all values were log$_2$-transformed. Median normalization across samples was performed (limma package) to reduce sample-level bias and TMT-batch effects. Differential protein abundance was assessed using the limma framework[78], which fits a linear model per protein and applies empirical Bayes variance moderation; significance was determined using a dual threshold of |log$_2$(fold change)| > 1 and $p$-value < 0.05 (static mode), with an alternative dynamic cutoff available based on distribution-derived thresholds. Results were visualized with EnhancedVolcano (version 1.14.0)[79].

## Hierarchical clustering and heatmap analysis of protein enrichment

Affinity-enriched proteins from cytosolic and nuclear fractions were analyzed under Mg$^{2+}$ or EDTA conditions in biological triplicates using TMT-based quantitative proteomics. A single TMT channel containing 33.3 μM compound was extracted across all multiplexes and experimental conditions to enable relative quantification. Protein-level reporter ion intensities were extracted and processed in R (v4.5.0), and proteins quantified in at least two out of three biological replicates in at least one condition were retained for analysis. Zero intensity values were treated as missing, and data were log$_2$-transformed prior to normalization. Log$_2$ intensities were median-normalized across samples using the limma package, and missing values were imputed using a minimal probability (MinProb) approach implemented in the imputeLCMD package, modeling low-abundance signals near the detection limit. Statistical significance was assessed using a three-way analysis of variance (ANOVA) testing the effects of compound, cellular fraction (cytosolic versus nuclear), and treatment (Mg$^{2+}$ versus EDTA), followed by Benjamini–Hochberg false discovery rate (FDR) correction. Proteins exhibiting a significant compound effect (FDR < 0.05) without significant contributions from fraction or treatment were selected, yielding 370 compound-responsive proteins, which were subsequently standardized by row-wise z-scoring and visualized by hierarchical clustering using the pheatmap package (v1.0.13)[80].

## Expression and purification of recombinant PMVK

The plasmid was designed using the GeneArt platform (Thermo Fisher Scientific). The PMVK coding sequence (UniProt: PMVK) was cloned into pET151/D-TOPO, encoding an N-terminal 6 × His–V5 tag and a TEV protease cleavage site under control of a T7 promoter. The construct was transformed into *E. coli* TOP10 cells, and plasmid DNA was isolated using the QIAprep Spin Miniprep Kit (Qiagen) according to the manufacturer's instructions. Correct insertion was confirmed by sequencing, and the verified plasmid was transformed into *E. coli* BL21 (DE3) cells.

Cells were grown in 50 mL LB medium supplemented with 8 g/l glycerol with ampicillin overnight at 37 °C until OD$_{600}$ reached ~5. The culture was then inoculated into 2 L fresh culture to 0.05 OD and grown for 2–3 h. Cultures were down-tempered to 18 °C over a period of 1 h before target expression was induced by the addition of 0.3 mM IPTG, and expression was allowed to continue for 22 h. Cells were harvested by centrifugation (5000 x $g$, 20 min, 4 °C), and the remaining pellet was resuspended in lysis buffer (20 mM Tris HCl, pH 8, 500 mM NaCl, 20 mM imidazole, 2 mM MgCl$_2$, 2 mM BME, 10% glycerol) (10 mL/g pellet) and 1 spatula tip of DNAseI, lysosyme, 1X cOmplete EDTA-free protease inhibitor was added upon thawing the pellet. Cells were lysed via a homogenizer (15000 psi, 5 iterations) and cell debris was removed by centrifugation (20,000 x $g$, 20 min, 4 °C). The supernatant was filtered through a 0.45 μm flask filter. The filtered lysate was loaded onto a Ni-charged HiTrap Chelating HP (GE Healthcare) column and washed with lysis buffer, followed by gradient elution with elution buffer 0%–100% (20 CV) (20 mM Tris HCl, pH 8, 500 mM NaCl, 500 mM imidazole, 2 mM MgCl$_2$, 2 mM BME, 10% glycerol). TEV cleavage was performed by the addition of TEV protease during dialysis buffer exchange in a Slide-A-Lyzer 10 000 MW cutoff dialysis container (dialysis against 2x 1 L Dialysis buffer (25 mM HEPES, pH 7, 250 mM NaCl, 0.5 mM DTT, 10% glycerol) for 12 h at 4 °C. The resulting solution was concentrated and loaded onto a Ni-charged HiTrap Chelating HP (GE Healthcare) column, and purified by reverse HiTrap (20 mM Tris HCl, pH 8, 500 mM NaCl, 20 mM imidazole, 2 mM MgCl$_2$, 2 mM BME, 10% glycerol).

## Isothermal Titration Calorimetry

PMVK was expressed and purified as previously described. Protein stocks were subjected to buffer exchange using final buffer conditions: 25 mM HEPES, pH 7.4, 150 mM KCl, 1 mM MgCl$_2$, 40 mM NaCl, 0.5 mM TCEP (ITC buffer). The exact protein concentration was determined by BCA assay and adjusted to 50 μM. InsP$_6$, 1PP-InsP$_5$, 5PP-InsP$_5$ and 1,5(PP)$_2$-InsP$_4$ were diluted to 500 μM in ITC buffer. ITC experiments were carried out at 25 °C in a MicroCal PEAQ-ITC calorimeter (Malvern Panalytical GmbH, Kassel, Germany), with ca. 50 μM protein in the cell and 500 μM ligand in the syringe. InsPs were titrated into the solution in nineteen 2 μL-steps. Spacing between injections was 150 s. The corresponding instrument software (MicroCal PEAQ-ITC Analysis) was used for baseline correction, peak integration, data fitting and determination of binding parameters. All experiments were carried out in triplicate.

## Reporting summary

Further information on research design is available in the Nature Portfolio Reporting Summary linked to this article.

## Data availability

The mass spectrometry raw data and ProteomeDiscoverer outputs generated in this study have been deposited in the ProteomeXchange consortium partner repository jPOSTrepo under accession codes JPST003145 [https://repository.jpostdb.org/preview/2101742205665782f5915de] and PXD052682. The MS processed datasets and chemical data generated in this study are provided in the Supplementary Information. Source data are provided with this paper. Source data are provided with this paper.

## Code availability

The R scripts used in this study are available without restrictions via Zenodo [https://zenodo.org/records/18375920]. Custom R scripts were developed by the authors with assistance from ChatGPT (OpenAI) for code optimization and troubleshooting. Final scripts were manually reviewed, tested, and validated.

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

## Acknowledgments

We thank Peter Schmieder for his 2D-NMR expertise and Lena von Oertzen for performing the cell culture experiments. We also thank Leonie Kurz for providing SYT1$^{C2B}$, and Meike Amma and Simon Bartsch for their help with the high-resolution mass spectrometry. We thank all group members of the Fiedler group for reading and discussing the manuscript.

## Author contributions

Conceptualization, A. Richter, J.A. Isern, D. Furkert, M. Ruwolt, and D. Fiedler; Methodology, A. Richter, J.A. Isern, D. Furkert, M. Ruwolt, S. Lampe, A. Majumdar, and D. Fiedler; Formal Analysis, A. Richter, J.A. Isern, M. Ruwolt, S. Lampe, and A. Majumdar; Investigation, A. Richter, J.A. Isern, M. Ruwolt, and S. Lampe; Writing – Original Draft, A. Richter, J.A. Isern D. Furkert and D. Fiedler; Writing – Review & Editing, J.A. Isern, and D. Fiedler; Visualization, A. Richter, J.A. Isern, M. Ruwolt, S. Lampe, and D. Fiedler; Supervision, F. Liu and D. Fiedler; Funding Acquisition, F. Liu and D. Fiedler.

## Funding

Annika Richter discloses support for the research of this work from the Studienstiftung des deutschen Volkes (German Academic Scholarship Foundation) and the Deutsche Forschungsgemeinschaft (DFG, German Research Foundation, project number 444048842). Jaime A. Isern discloses support for the research of this work from the Swiss National Science Foundation (CRSII5_20941). Open Access funding enabled and organized by Projekt DEAL.

## Competing interests

F. Liu is a shareholder and advisory board member of Absea Biotechnology Ltd and VantAI. The remaining authors declare no competing interests.
