## [Transparent Peer Review file · Nature Communications]

Proteome-wide quantification of inositol pyrophosphate-protein interactions

Corresponding Author: Professor Dorothea Fiedler

Version 0:

Reviewer comments:

Reviewer #1

(Remarks to the Author)

In this study, the authors have developed novel tools that can be valuable for researchers investigating polyphosphoinositols, particularly those from the pyrophosphate class. These tools have the potential to advance our understanding of polyphosphoinositol functions and interactions.

The authors employed these novel tools to affinity purify potential binding proteins from both cell and nuclear lysates. By using varying concentrations of ligand bound to the affinity column, they attempted to approximate the binding affinities of the proteins for the specific ligand. This approach aims to provide a more realistic estimation of these interactions, which could be beneficial for future research.

Strengths

The primary strength of this paper lies in the preliminary application of the newly developed tools. The results presented are of high quality and demonstrate the potential utility of these tools for identifying polyphosphoinositol-binding proteins.

Major Issues:

1. Endogenous Ligand Interaction:

It is not clear whether the authors have demonstrated that any of the binding proteins identified are capable of interacting with endogenous, non-chemically modified ligands. This is a critical point, as the biological relevance of these interactions depends on their occurrence with naturally occurring ligands. Moreover deriving affinities based on column bound amounts may not really reflect interactions with soluble ligands. the authors should perform binding assays for some identified proteins using soluble inositol phosphates and demonstrate similar affinities. moreover these column are highly negatively charged and the worry would be that the interaction is solely charge based rather than representing a true interaction site. In these studies what was the control column that was utilised? did it have negative charges? if not then the authors should include a comparison with a strongly anionic charged column and determine if similar proteins interact and how they are displaced by elution with the various specific pyrophosphorylated inositols.

2. **Functional Relevance in Signaling Pathways:**

The authors need to show that some of the newly identified proteins play a functional role in the signaling pathways mentioned. Specifically, the study would benefit from experiments that involve genetically manipulating enzymes within those pathways to demonstrate that the identified proteins are indeed regulated in a biologically meaningful way.

3. **Relevance of Final Results (Figure 6):**

The significance of the findings presented in Figure 6 remains unclear. It is important to determine whether there is a statistically significant enrichment of pyrophosphorylated proteins. Additionally, the authors should use this dataset to confirm whether some of the novel proteins identified are indeed pyrophosphorylated, as this would strengthen the biological relevance of their findings.

the work is significant for the field and is original and if the authors can provide evidence of relevance of some of these proteins this would provide impetus to others to investigate their favourite protein and how it may function within these pathways.

(Remarks on code availability)

Reviewer #2

(Remarks to the Author)

Key results

The authors synthesized four biotinylated affinity probes of InsPs and PP-InsPs (5PP-InsP5, 1PP-InsP5, 1,5(PP)2-InsP4, and InsP6) and used these to determine proteome-wide apparent binding constants. Using western blot the authors determined that these probes were able to enrich known PP-InsP binding proteins, as well as demonstrating that these proteins could be enriched in a concentration-dependent manner. Furthermore, the presence of Mg²⁺ ions was shown to significantly affect the binding interactions between PP-InsPs and their interactors. The authors determined the apparent binding constants for a wide range of proteins with values ranging from nanomolar to micromolar scale, in the presence of either EDTA or Mg²⁺ ions. Over 1000 proteins were found to interact with their probes, with a particular focus on the interactors binding with apparent binding constants below 5 μM. 1,5(PP)2-InsP4 and 5PP-InsP5 exhibited strong interactions with proteins involved in RNA processing, ribosome biogenesis, and other nuclear functions. A subset of PP-InsP interactors were linked to pyrophosphorylation. The authors validated four proteins as novel targets for pyrophosphorylation. Finally, they determined that compared to other PP-InsPs, 1PP-InsP5 displayed weaker interactions compared to the KdApps linked to the other synthesized probes. The authors suggest that this PP-InsP might primarily function as an intermediate in PP-InsP turnover rather than being a significant signaling molecule.

Validity

- The use of a mass spectrometry-based approach to determine protein-(PP)-InsP apparent binding affinities provides a precise and high-throughput method for analyzing these interactions. The careful design of the biotinylated affinity reagents, which were validated against their known interactors, adds credibility to their methodology. However, the impact of the addition of a biotin group onto these molecules is not discussed in this paper. The linked biotin group could for example enrich their own set of interactors which might not be necessarily related to (PP)-InsP functioning or affect the apparent binding constants.
- The authors thoroughly examined the role of Mg²⁺ ions in modulating (PP)-InsP-protein interactions and provided valuable insights into the variation of binding affinities among various (PP)-InsPs. Nonetheless, the conclusion drawn that 1PP-InsP5 is a less dominant signaling molecule based on the weaker observed binding affinities, would benefit from further validation through additional functional studies.
- Additionally, while the identification of potential pyrophosphorylation targets is a valuable addition to this paper, the study falls short of fully validating the functional relevance of these proteins in cellular signaling, leaving this aspect open for future investigation.
- The authors have profiled apparent affinities but have not performed relative quantification experiments of all the baits against each other to further illustrate binding specificity. Such results could be visualized in a hierarchical clustering figure, for example
- It seems that the authors did not perform and biological replicate analysis for their mass spec measurements
- Figure 3 (western blot) lacks negative controls

Significance

- The application of the method to identify proteome-wide apparent binding affinities by mass spectrometry in protein extracts provides new information for the field of (PP)-InsP interacting proteins. The authors contribute valuable data to the field. This adaptation highlights the method's potential applicability in other areas of research involving small molecule-protein interactions. Furthermore, understanding how (PP)-InsPs interact with specific proteins, particularly in the context of RNA processing and ribosome biogenesis, can pave the way for new experimental studies to investigate the regulation of gene expression and translation.
- Significance of the current manuscript is significantly compromised by a previous PIP interaction proteomics survey that is not cited or discussed in the current manuscript (PMID: 24462288). Additional PIP interaction screenings have also been performed, these have also not been discussed

Suggested additional improvements

- The paper would benefit from additional controls investigating how the biotin group affects the interactions between the affinity probes and their interactors. For example, a synthesized version of these affinity probes lacking the biotin group would be appropriate in a competitive binding assay, to determine for a select group of proteins that they indeed solely bind to the (PP)-InsP part of the affinity probe.
- The suggestion of the authors that 1PP-InsP5 is primarily an intermediate, could be further backed up by a couple of approaches. They could for example determine its subcellular location in the cell, by using a fluorescently tagged analogue of 1PP-InsP5. They could also determine the potential turnover rate of this ligand and comparing that to the other PP-InsPs.
- Relative quantification experiments of all PIP baits against each other should be performed to further illustrate binding specificity.
- Lastly, functional assays to explore the role of identified pyrophosphorylation targets and the biological relevance of PP-

InsP interaction would greatly enhance the impact of this study.

Clarity and context

- The introduction would benefit from improving the flow to increase understanding of the reader of the paper objectives and necessary background information. The introduction jumps relatively quickly into specifics of InsPs and PP-InsPs for example, which could also be mentioned later in the text after the general roles of these molecules are made clear. Furthermore, the broader implications of the study's findings could be more clearly stated.
- The results section seems to be clearly structured, with preliminary and validation experiments presented first followed by hypothesis-driven analysis (identification of pyrophosphorylated proteins). Key results are clearly summarized.
- While the flow of the discussion section is generally strong, it would benefit from a better connection between certain results and their implications, clearly stating what the significance is of these findings.

(Remarks on code availability)

Reviewer #3

(Remarks to the Author)

A high quality paper that uses new inositol poly/pyro-phosphate tool compounds to identify inositol poly/pyro-phosphate binding proteins in mammalian systems. The approach captured the apparent binding constants of interacting proteins at a proteome scale. This work identified many known and previously unidentified interacting proteins and provided information about their selectivity, apparent affinity and Mg²⁺ dependence. Data analysis and further experiments identified several new pyro-phosphorylated proteins.

The synthetic chemistry involved in making the inositol poly/pyro-phosphates tool compounds is high quality and extremely well documented and validated. The data describing the identification and characterisation of the inositol poly/pyro-phosphate binding proteins and pyro-phosphorylated proteins will, as the authors argue, be a useful resource for the field. That there are apparently relatively few "high"-affinity 1-PP InsP5 -selective binding proteins is striking and interesting. The impact of Mg²⁺ on binding (selectivity and apparent affinities) is clearly validated and again an interesting feature of the results, although somewhat expected on the basis of previous work (that is cited).

Overall the MS is very well presented and written.

I have a few relatively minor questions/ comments I believe the authors should address.

It wasn't clear to me how the "concentration" of the inositol phosphate tools once bound to the sepharose beads (incidentally I couldn't find the source of the streptavidin beads used) were calculated for the purposes of estimating apparent affinities. I assumed it was the amount known to be bound to the beads over the total aqueous volume with the beads. It is possible to argue it is more appropriate to use the volume of beads as that is space in which ligands are presented.

It would be helpful to know what typical proportions of proteins remained on the beads after elution with excess competing ligands.

It would also be helpful to know how the modified , metabolically resistant pyro-phosphate analogues compared to the natural species in their ability to compete, giving an indication of the extent to which the modifications impacts binding.

In the text the authors refer to 11plex TMT labelling yet in the methods section in the critical commercial reagents section it appears a 10-plex was used. Was this a 10-plex combined with use of a separate 1-plex kit?

The study does have some very useful comparisons between tool compounds that afford a good view of the selectivity of binding, however, it would be very useful for the field and this study if a non-biological enantiomer of either 1PP (3PP) or 1,5PP was available. The chemistry did yield some very nice enantiomerically pure intermediates that would support this approach but it would be an unreasonable request if the relevant enantiomers were not already or easily derived.

In my opinion the main issue with this paper for Nat Comms is whether the advance/resource it offers is significant enough to merit potential publication. For me, big advances for the field would for example include identification of new inositol poly/pyro-phosphate binding proteins and establishing they are subject to changes in properties associated with binding under physiological conditions. Especially if there was evidence the regulation was dynamic under physiological conditions. For example that changed levels of inositol poly/pyro-phosphate changed the activity/distribution/abundance of a relevant binding protein.

(Remarks on code availability)

Version 1:

Reviewer comments:

Reviewer #1

(Remarks to the Author)

I thank the authors for their detailed rebuttal and for the additional experiments that have been included in the revised manuscript. Overall, the responses to most of my comments (1 and 3) are fine and the additional data strengthen several aspects of the study.

However, with respect to major point 2, I remain somewhat unconvinced by the authors' response.

The authors have now provided additional biochemical evidence demonstrating binding of InsP6/PP-InsPs to several newly identified targets and have included direct affinity measurements (e.g., for PMVK using ITC), as well as an example showing that these messenger molecules can modulate the enzymatic activity of TNIK. These additions are valuable and certainly improve the manuscript.

That said, the central concern raised in my original comment was whether any of the newly identified binding proteins functionally participate in the signaling pathways suggested by the proteomic analysis. The experiments presented primarily demonstrate biochemical interactions rather than functional relevance within a cellular context. As such, the manuscript still reads largely as a methodological and resource paper, focused on the generation and validation of affinity reagents and the resulting proteomic datasets.

For a journal of this scope, I would ideally expect at least some initial functional validation demonstrating that perturbation of the relevant metabolic pathway (for example, through genetic manipulation of enzymes controlling PP-InsP levels) leads to measurable regulation of one or more of the identified targets. Even limited cell-based validation would significantly strengthen the biological impact of the work.

I understand the authors' argument that comprehensive functional studies on multiple targets could take substantial time and may fall outside the immediate scope of the manuscript. Nevertheless, some demonstration of pathway relevance would help move the study beyond a primarily methodological framework.

Given this, I will leave it to the editors to decide whether the current level of biological validation is sufficient for publication in this journal

(Remarks on code availability)

Reviewer #2

(Remarks to the Author)

The authors have addressed my main concerns in a satisfactory manner.

(Remarks on code availability)

“Proteome-wide quantification of inositol pyrophosphate-protein interactions”

Point-by-point response to reviewers

Reviewer 1

In this study, the authors have developed novel tools that can be valuable for researchers investigating polyphosphoinositols, particularly those from the pyrophosphate class. These tools have the potential to advance our understanding of polyphosphoinositol functions and interactions.

The authors employed these novel tools to affinity purify potential binding proteins from both cell and nuclear lysates. By using varying concentrations of ligand bound to the affinity column, they attempted to approximate the binding affinities of the proteins for the specific ligand. This approach aims to provide a more realistic estimation of these interactions, which could be beneficial for future research.

Strengths

The primary strength of this paper lies in the preliminary application of the newly developed tools.

The results presented are of high quality and demonstrate the potential utility of these tools for identifying polyphosphoinositol-binding proteins.

Major Issues:

1. Endogenous Ligand Interaction:

It is not clear whether the authors have demonstrated that any of the binding proteins identified are capable of interacting with endogenous, non-chemically modified ligands. This is a critical point, as the biological relevance of these interactions depends on their occurrence with naturally occurring ligands.

We thank the reviewer for raising this important point. We have now addressed this point in the section “A proteomics workflow for determination of apparent binding constants” (lines 289–318). Specifically, we performed competition assays in which increasing concentrations of the non-modified, endogenous ligands were added prior to enrichment with the corresponding affinity reagents. These experiments were conducted for multiple ligands and representative targets, including PTK2, NAMPT, STK24, PRPS1, DIPP1, and PMVK, and were visualized by Western blot (New Figure 4b). Across targets, ligands, and affinity reagents, we observed clear and dose-dependent competition profiles, strongly suggesting that endogenous ligands and the corresponding affinity reagents compete for the same binding

sites. In addition, an experiment using recombinant XPR1^{SPX} further confirms direct competition between InsP₆ and immobilized b-InsP₆ (New Supplementary Figure S6).

To further validate InsP/PP-InsP interactions, we performed isothermal titration calorimetry (ITC) to determine the dissociation constants of recombinantly expressed PMVK with InsP₆, 5PP-InsP₅, and 1,5(PP)₂-InsP₄. Among the ligands tested, InsP₆ displayed the highest affinity, with progressively weaker binding observed for the more highly charged PP-InsPs (New Figure 4c and New Supplementary Figure S12). Lastly, to evaluate whether InsP₆/PP-InsPs influence enzymatic activity, we assayed six recombinantly expressed enzymes (TNIK, PMVK, STK24, PTK2, NAMPT, and PRPS1) in the presence of InsP₆, 5PP-InsP₅, and 1,5(PP)₂-InsP₄. Of the kinases tested, TNIK (TRAF2- and NCK-interacting kinase) exhibited a concentration-dependent reduction in activity upon addition of PP-InsPs (New Figure 4d). Together, these results establish that the identified proteins interact with naturally occurring, non-chemically modified inositol (pyro)phosphates, supporting the biological relevance of the interactions captured by the proteomics experiments.

Moreover, deriving affinities based on column bound amounts may not really reflect interactions with soluble ligands. the authors should perform binding assays for some identified proteins using soluble inositol phosphates and demonstrate similar affinities.

We agree with the reviewer, that the K_D^{app} values derived from the resin bound affinity reagents cannot be directly translated to actual binding constants with the free ligands. In the experimental set-up for the proteomics experiments we are not at equilibrium, and the ligands are modified. Therefore, the derived K_D^{app} values always need to be taken with caution. We have added this cautionary note to the discussion (lines 543-546). That being said, the K_D^{app} values for several of the known PP-InsP binding proteins, such as PPIP5K2 or DIPPI1, are on the same order of magnitude as the binding constants for the native ligands. A systematic comparison using ITC would, in our opinion, not necessarily provide a clearer picture, as recombinant proteins are often not adequately post-translationally modified, may not have been folded properly, and may lack other protein binding partners that are relevant for the interaction with the ligands. We therefore hope that the reviewer is satisfied with the critical discussion we have added (lines 306-312, 543-546, 554-561).

Moreover, these columns are highly negatively charged and the worry would be that the interaction is solely charge based rather than representing a true interaction site. In these studies what was the control column that was utilised? did it have negative charges? if not then the authors should include a comparison with a strongly anionic charged column and determine if similar proteins interact and how they are displaced by elution with the various specific pyrophosphorylated inositols.

“Authors should include a comparison with a strongly anionic charged column and determine if similar proteins interact and how they are displaced by elution with the various specific pyrophosphorylated inositols”

We thank the reviewer for his/her thoughts regarding the potential contribution of nonspecific electrostatic interactions. Of course, electrostatic forces are a major contributing factor in the interactions of InsP₆/PP-InsPs with protein binding partners. Many crystal structures illustrate how the highly negatively charged molecules bind to highly positively charged regions of proteins. However, several lines of evidence indicate that there is specificity in the observed interactions.

First, distinct apparent dissociation constants (K_D^{app}) were obtained for different proteins and ligands, suggesting that the recognition of the ligands goes beyond plain electrostatics. This view is further supported by the newly added competition assays described in “A proteomics workflow for determination of apparent binding constants” (lines 289-305, New figure 4b). If binding were driven exclusively by charge, comparable competition profiles would be expected across ligands and targets; however, we observe ligand-, target-, and probe-dependent displacement behavior. Second, several controls were included in the proteomics experiments, namely no-probe controls as well as probe-plus-competitor conditions (Supplementary Figure S6). These controls allow us to distinguish probe-dependent enrichment from nonspecific binding to the matrix. In addition, hierarchical clustering analysis (Supplementary Figure S11) and probe-to-probe comparative quantification (Supplementary Figure S12), conducted for one probe concentration across all conditions, revealed specificity patterns that are inconsistent with simple charge-based interactions (lines 319-331). Marked differences in protein enrichment were observed between chemically distinct probes carrying the same net charge (e.g., b-1PCP-InsP₅ vs. b-5PCP-InsP₅), further supporting the conclusion that these recognition processes go beyond plain electrostatic attractions in these assays. Because we can make these comparisons between the InsP₆/PP-InsP ligands, we feel that a generic strongly anionic column would not provide additional mechanistic insights.

2. ****Functional Relevance in Signaling Pathways:****

The authors need to show that some of the newly identified proteins play a functional role in the signaling pathways mentioned. Specifically, the study would benefit from experiments that involve genetically manipulating enzymes within those pathways to demonstrate that the identified proteins are indeed regulated in a biologically meaningful way.

We appreciate the reviewer’s comment. We have now included additional experiments, demonstrating that InsP₆/PP-InsPs bind with distinct affinities to several newly identified targets, including DIPP1, PRPS1, PTK2, NAMPT, STK24, and PMVK (lines 289-305; new Figure 4b). We further characterized the newly identified protein–ligand interactions by directly measuring binding affinities of InsP₆/PP-InsPs to PMVK using ITC (lines 306-311;

Figure 4c; Supplementary Figure S10). Moreover, we show in biochemical experiments, that these messenger molecules can modulate the enzymatic activity of the kinase TNIK (lines 311-316; Figure 4d).

Our primary goal in this study was to first of all synthesize affinity reagents for the PP-InsPs, as this had not been done for 1PP-InsP₅ and 1,5(PP)₂-InsP₄ in the past. We then thoroughly validate these reagents, and finally demonstrate their applicability in proteomics experiments, in particular with regards to deriving K_D^{app} values on a proteome wide scale. As part of the revision process, we have reprocessed all of the proteomics data with more stringent cut-offs, so that we can be confident in our functional analysis (line 270-277). The overarching trends in these proteomics data sets revealed a strong influence of Mg²⁺ ions for mediating certain protein-ligand interactions and a connection to protein pyrophosphorylation. We subsequently followed up on putative protein targets of pyrophosphorylation, and could identify several new modification sites.

We envision that researchers from different disciplines can now use our data sets as resources and conduct the functional studies the reviewer suggests for their proteins of interest. To include such an added investigation into the current manuscript goes beyond the scope of our work, as it could take years to establish the relevant cell-based model systems and assays.

3. ****Relevance of Final Results (Figure 6):****

The significance of the findings presented in Figure 6 remains unclear. It is important to determine whether there is a statistically significant enrichment of pyrophosphorylated proteins. Additionally, the authors should use this dataset to confirm whether some of the novel proteins identified are indeed pyrophosphorylated, as this would strengthen the biological relevance of their findings.

We thank the reviewer for this comment regarding the interpretation of Figure 6 (now Figure 7). We have clarified the significance of these results in the revised manuscript in the section “Enriched proteins can be linked to pyrophosphorylation-based signal transduction” (lines 483-513). In this section, we now explicitly describe the enrichment analysis and its statistical evaluation, demonstrating a striking enrichment of known pyrophosphorylated proteins within the datasets. Among the 603 Mg²⁺-dependent PP-InsP₅ and 1,5(PP)₂-InsP₄ binders, 24 were known pyrophosphoproteins. Currently, there are 59 characterized pyrophosphoproteins in HEK293T cells. Therefore, this enrichment of pyrophosphoproteins is highly significant, $p = 7.7 \times 10^{-22}$). We then wondered if we could exploit this enrichment to identify additional pyrophosphorylation sites. Based on known features of protein pyrophosphorylation - such as priming phosphorylation by acidophilic kinases, and localization to highly disordered regions - we narrowed down the number of putative pyrophosphorylation targets among the ca. 600 interactors. For several of these putative targets, we had inconclusive MS data from past experiments. Using the affinity-enrichment data as an “indicator” of protein pyrophosphorylation, we then specifically assayed the putative modification sites in a

targeted MS approach and, excitingly, could indeed confirm several of them. Our affinity enrichment data therefore seems to provide a springboard to further investigate protein pyrophosphorylation. One may even consider the affinity reagents described here for the specific enrichment of pyrophosphoproteins in future proteomics workflows.

The work is significant for the field and is original and if the authors can provide evidence of relevance of some of these proteins this would provide impetus to others to investigate their favourite protein and how it may function within these pathways.

We thank the reviewer for recognizing the originality and significance of this work. By thoroughly revising the original manuscript, by providing many additional validation experiments, by including biophysical and biochemical validation of binding interactions, and by reprocessing all of the proteomics data and subsequent analyses, we think we have conclusively demonstrated that InsP₆/PP-InsPs engage selected targets through distinct modes of action. Our results provide concrete examples of biological relevance and illustrate how the chemical tools and proteomic datasets presented here can be used to interrogate individual proteins of interest. We are therefore confident that future studies will build on the data and the framework established here - by us and many others in the field.

Reviewer #2

Key results

The authors synthesized four biotinylated affinity probes of InsPs and PP-InsPs (5PP-InsP5, 1PP-InsP5, 1,5(PP)2-InsP4, and InsP6) and used these to determine proteome-wide apparent binding constants. Using western blot the authors determined that these probes were able to enrich known PP-InsP binding proteins, as well as demonstrating that these proteins could be enriched in a concentration-dependent manner. Furthermore, the presence of Mg²⁺ ions was shown to significantly affect the binding interactions between PP-InsPs and their interactors. The authors determined the apparent binding constants for a wide range of proteins with values ranging from nanomolar to micromolar scale, in the presence of either EDTA or Mg²⁺ ions. Over 1000 proteins were found to interact with their probes, with a particular focus on the interactors binding with apparent binding constants below 5 μ M. 1,5(PP)2-InsP4 and 5PP-InsP5 exhibited strong interactions with proteins involved in RNA processing, ribosome biogenesis, and other nuclear functions. A subset of PP-InsP interactors were linked to pyrophosphorylation. The authors validated four proteins as novel targets for pyrophosphorylation. Finally, they determined that compared to other PP-InsPs, 1PP-InsP5 displayed weaker interactions compared to the KdApps linked to the other synthesized probes. The authors suggest that this PP-InsP might primarily function as an intermediate in PP-InsP turnover rather than being a significant signaling molecule.

Validity

- The use of a mass spectrometry-based approach to determine protein-(PP)-InsP apparent binding affinities provides a precise and high-throughput method for analyzing these interactions. The careful design of the biotinylated affinity reagents, which were validated against their known interactors, adds credibility to their methodology. However, the impact of the addition of a biotin group onto these molecules is not discussed in this paper. The linked biotin group could for example enrich their own set of interactors which might not be necessarily related to (PP)-InsP functioning or affect the apparent binding constants.

We thank the reviewer for making this point. In our workflow for affinity enrichment, the biotin moiety serves exclusively as a tag for immobilization, and the biotinylated molecules are first immobilized on streptavidin-agarose beads, before being exposed to cell lysates. The biotin-streptavidin interaction is among the tightest known protein-ligand interactions, and proteins from our cell lysates will not compete with, or displace, biotin from streptavidin under the experimental conditions used. Consequently, the affinity enrichment experiments are driven by interactions between the immobilized (PP)-InsP probe and cellular proteins, rather than by direct interactions with the biotin group itself. To clarify that the biotinylated affinity reagents are immobilized to the beads prior to lysate treatment, we have adjusted the wording throughout the manuscript to read “immobilized biotin-InsP₆/PCP-InsP” or “resin-bound biotin InsP₆/PCP-InsPs”.

Nevertheless, we further validated, that the captured proteome reflects binding interactions with InsP₆/PCP-InsPs, and not with the streptavidin beads or the biotin itself. Specifically, we performed competition assays in which increasing concentrations of the non-modified, endogenous ligands were added prior to enrichment with the corresponding affinity probes (lines 289-305). These experiments were conducted for multiple ligands and representative targets, including PTK2, NAMPT, STK24, PRPS1, DIPP1, and PMVK, and were visualized by Western blot (New Figure 4b). Across targets, ligands, and affinity reagents, we observed clear and dose-dependent competition profiles, strongly suggesting that endogenous ligands and the corresponding affinity reagents compete for the same binding sites. In addition, an experiment using recombinant XPR1^{SPX} further confirms direct competition between InsP₆ and immobilized b-InsP₆ (New Supplementary Figure S6).

Based on these considerations and results, we believe that the biotin tag does not introduce a significant bias in target enrichment.

- The authors thoroughly examined the role of Mg²⁺ ions in modulating (PP)-InsP-protein interactions and provided valuable insights into the variation of binding affinities among various (PP)-InsPs. Nonetheless, the conclusion drawn that 1PP-InsP5 is a less dominant signaling molecule based on the weaker observed binding affinities, would benefit from further validation through additional functional studies.

We appreciate the reviewer's comment and agree that this hypothesis may be premature. Since it was not our goal to determine, which PP-InsP is "the most important one", or the physiologically most relevant one, we have now deleted this section from both the Results and the Discussion sections.

- Additionally, while the identification of potential pyrophosphorylation targets is a valuable addition to this paper, the study falls short of fully validating the functional relevance of these proteins in cellular signaling, leaving this aspect open for future investigation.

We thank the reviewer for this insightful comment. We were positively surprised to observe such a strong enrichment of pyrophosphorylated proteins with our affinity reagents. Being able to source this data to then identify additional modification sites was a useful application of this data, since the identification of pyrophosphorylation sites remains a difficult task. To date, the most comprehensive analysis of protein pyrophosphorylation encompasses ca. 150 modification sites (Morgan et al., Nat Chem Biol, 2024).

It is certainly our long-term goal to validate the functional relevance of these modification sites, but the current methods are limited. Validating the functional relevance of individual phosphorylation sites can already be challenging, although phosphorylation has been studied for decades. Protein pyrophosphorylation is a more recent topic, and hence the tools and methods are still being developed (in part by our lab). Introducing a loss of function

mutation, by mutating the modification site to alanine (as is often done to investigate protein phosphorylation) will abrogate both phosphorylation and pyrophosphorylation. Delineating the contributions of the priming kinase and the PP-InsPs therefore already becomes difficult. Also, Glu or Asp are not suitable for mimicking a potential gain of function mutation, as they will more closely resemble a phosphorylation rather than a pyrophosphorylation site. And finally, pyrophosphorylation sites often localize to highly disordered and densely modified regions of proteins. Pinpointing specific effects of one pyrophosphorylation site in the context of other nearby phosphorylation or pyrophosphorylation sites is therefore challenging. We are eager to tackle these questions in the future, but given the technical limitations, we feel that these investigations will be very time consuming and risky at the current time.

- The authors have profiled apparent affinities but have not performed relative quantification experiments of all the baits against each other to further illustrate binding specificity. Such results could be visualized in a hierarchical clustering figure, for example.

We thank the reviewer for this constructive suggestion. We have taken up this suggestion and the data is included and discussed in the revised manuscript (lines 319-331, New Supplementary Figures S11 and S12). Probe-to-probe comparative quantification, conducted for one probe concentration across all conditions, followed by hierarchical clustering analysis, revealed interesting specificity patterns. Marked differences in protein enrichment were observed between the chemically distinct probes, the different conditions (with and without Mg^{2+} ions), and the distinct cellular fractions.

- It seems that the authors did not perform and biological replicate analysis for their mass spec measurements

We thank the reviewer for this comment. All the TMT based proteomic experiments were conducted in biological triplicates. This has been clarified both in the manuscript, as well as in the materials and methods section.

- Figure 3 (western blot) lacks negative controls

In Figure 3a, the lysate lane serves as a positive control, confirming the presence of the target proteins in the input material. As these experiments assess binding capacity, comparison between the supernatant and elution fractions provides an internal control for reagent-dependent enrichment.

In addition, the Ctrl lanes serve as negative controls: without immobilized reagents, a band is expected in the supernatant, whereas no band should be present in the eluted fractions. In Figure 3c, the lysate again serves as a positive control (presence of the target proteins in

the cell extract), while the 0 μM affinity reagent condition (i.e., no affinity reagent added) functions as a negative control (no enrichment in absence of affinity reagent). We have revised the figure caption to explicitly state the identity and role of these controls to avoid ambiguity (lines 229, 239-240) and thank the reviewer for this comment.

Significance

- The application of the method to identify proteome-wide apparent binding affinities by mass spectrometry in protein extracts provides new information for the field of (PP)-InsP interacting proteins. The authors contribute valuable data to the field. This adaptation highlights the method's potential applicability in other areas of research involving small molecule-protein interactions. Furthermore, understanding how (PP)-InsPs interact with specific proteins, particularly in the context of RNA processing and ribosome biogenesis, can pave the way for new experimental studies to investigate the regulation of gene expression and translation.

We thank the reviewer for this positive assessment of our work.

- Significance of the current manuscript is significantly compromised by a previous PIP interaction proteomics survey that is not cited or discussed in the current manuscript (PMID: 24462288). Additional PIP interaction screenings have also been performed, these have also not been discussed

We appreciate the reviewer's observation. However, we do not consider our work to be compromised by the study of Jungmichel et al. ("Specificity and Commonality of the Phosphoinositide-Binding Proteome Analyzed by Quantitative Mass Spectrometry", PMID: 24462288). The cited study focuses on profiling the phosphoinositide (PIP) interactome using affinity reagents based on lipid-anchored phosphatidylinositols with a low degree of phosphorylation (one to three phosphoryl groups). By contrast, our study focusses on the non-lipidated, soluble, very densely phosphorylated messengers InsP₆/PP-InsPs (six to eight phosphoryl groups).

Both the lipid-anchored phosphatidylinositols (PIPs) and inositol poly/pyrophosphates (InsPs/PP-InsPs) are important signaling molecules, yet they represent distinct messenger families as the PIPs are membrane-anchored and the InsPs/PP-InsPs are freely diffusible. Nevertheless, they can have overlapping functions. For example, we previously demonstrated that among proteins that bound to InsP₆ and 5PP-InsP₅, PH- and C2-domain-containing proteins were overrepresented (Furkert et al., PMID: 32783964). Also in our current data sets, a Pfam domain analysis of all identified targets with K_D^{app} values $< 5 \mu\text{M}$ revealed an enrichment of PH, PX, C1, C2, FERM, and ENTH-domain containing proteins. This is consistent with previous studies that showed that InsP₆/5PP-InsP₅ can indeed target the same binding sites as PIPs. This is also consistent with the proposal that one mechanism of action for InsPs/PP-InsPs is to compete for these PIP-binding domains. If the reviewer

deems it necessary to include the Pfam analysis in the current manuscript, we would be happy to do so. However, we felt that such an added section would not contribute to the overall story of the manuscript as it was not our goal to elucidate the cross-talk between these two groups of distinct messenger molecules. Instead, with our work we wish to disclose a comprehensive data set, that will capture all InsP₆/PP-InsP interacting partners, regardless of their mode of action (whether this is competition with PIPs, serving as molecular glue, or protein pyrophosphorylation).

Therefore, we believe there is no conflict between the two studies; rather, they are complementary: Our study introduced biotinylated InsP/PCP-InsPs affinity reagents with higher phosphorylation states (six, seven, and eight), enabling the characterization of interactomes specific to inositol polyphosphate and pyrophosphate signaling species that could not be addressed in PIP-based screens.

Suggested additional improvements

- The paper would benefit from additional controls investigating how the biotin group affects the interactions between the affinity probes and their interactors. For example, a synthesized version of these affinity probes lacking the biotin group would be appropriate in a competitive binding assay, to determine for a select group of proteins that they indeed solely bind to the (PP)-InsP part of the affinity probe.

We appreciate this suggestion. To address this, we have added further validation experiments (lines 201-204, 217-219, Supplementary Figure S6). In addition, in the section “A proteomics workflow for determination of apparent binding constants” we now include competition assays in which increasing concentrations of the non-modified, endogenous ligands were added prior to enrichment with the corresponding affinity probes. These experiments were conducted for multiple ligands and representative targets, including PTK2, NAMPT, STK24, PRPS1, DIPP1, and PMVK, and were visualized by Western blot (lines 289-305, New Figure 4b). Across targets, ligands, and affinity reagents, we observed clear and dose-dependent competition profiles, strongly suggesting that endogenous ligands and the corresponding affinity reagents indeed compete for the same binding sites.

- The suggestion of the authors that 1PP-InsP₅ is primarily an intermediate, could be further backed up by a couple of approaches. They could for example determine its subcellular location in the cell, by using a fluorescently tagged analogue of 1PP-InsP₅. They could also determine the potential turnover rate of this ligand and comparing that to the other PP-InsPs.

We thank the reviewer for this thought. We agree that determining the subcellular localization and turnover rate of 1PP-InsP₅ would provide valuable insight into its potential biological role. However, we believe that validating this hypothesis is out of the scope of the current

manuscript. Therefore, to avoid overinterpretation, we have revised the manuscript and deleted this section from the Results and Discussion sections.

- Relative quantification experiments of all PIP baits against each other should be performed to further illustrate binding specificity.

We thank the reviewer for this constructive suggestion. As mentioned above, we have now added a comparative relative quantification of the InsP₆/PP-InsP reagents in the revised manuscript (lines 319-331, New supplementary figures S11 and S12).

- Lastly, functional assays to explore the role of identified pyrophosphorylation targets and the biological relevance of PP-InsP interaction would greatly enhance the impact of this study.

We thank the reviewer for this comment. It is certainly our long-term goal to validate the functional relevance of pyrophosphorylation sites, but the current methods are, unfortunately, limited. Validating the functional relevance of individual phosphorylation sites can already be challenging, although phosphorylation has been studied for decades. Protein pyrophosphorylation is a more recent topic, and hence the tools and methods are still being developed (in part by our group). Introducing a loss of function mutation, by mutating the modification site to alanine (as is often done to investigate protein phosphorylation) will abrogate both phosphorylation and pyrophosphorylation. Delineating the contributions of the priming kinase and the PP-InsPs therefore already becomes difficult. Also, Glu or Asp are not suitable for mimicking a potential gain of function mutation, as they will closely resemble a phosphorylation rather than a pyrophosphorylation. And finally, pyrophosphorylation sites often localize to highly disordered and densely modified regions of proteins. Pinpointing specific effects of one pyrophosphorylation site in the context of other nearby phosphorylation or pyrophosphorylation sites is therefore challenging. We are eager to tackle these questions in the future, but given the technical limitations, we feel that these investigations will be very time consuming and risky at the current time.

Clarity and context

- The introduction would benefit from improving the flow to increase understanding of the reader of the paper objectives and necessary background information. The introduction jumps relatively quickly into specifics of InsPs and PP-InsPs for example, which could also be mentioned later in the text after the general roles of these molecules are made clear. Furthermore, the broader implications of the study's findings could be more clearly stated.

- The results section seems to be clearly structured, with preliminary and validation experiments presented first followed by hypothesis-driven analysis (identification of pyrophosphorylated proteins). Key results are clearly summarized.

- While the flow of the discussion section is generally strong, it would benefit from a better connection between certain results and their implications, clearly stating what the significance is of these findings.

We appreciate the reviewer's suggestions. We have revised the Introduction and Discussion sections to improve overall flow and clarity. Additional background information was included to better guide the reader through the study's objectives, and we tried more clearly articulate the broader implications and significance of our findings (lines 37, 42-44, 49-51, 105-118, 533-549, 554-561, 630-638).

Reviewer #3

A high-quality paper that uses new inositol poly/pyro-phosphate tool compounds to identify inositol poly/pyro-phosphate binding proteins in mammalian systems. The approach captured the apparent binding constants of interacting proteins at a proteome scale. This work identified many known and previously unidentified interacting proteins and provided information about their selectivity, apparent affinity and Mg²⁺ dependence. Data analysis and further experiments identified several new pyro-phosphorylated proteins.

The synthetic chemistry involved in making the inositol poly/pyro-phosphates tool compounds is high quality and extremely well documented and validated. The data describing the identification and characterisation of the inositol poly/pyro-phosphate binding proteins and pyro-phosphorylated proteins will, as the authors argue, be a useful resource for the field. That there are apparently relatively few "high"-affinity 1-PP InsP₅ -selective binding proteins is striking and interesting. The impact of Mg²⁺ on binding (selectivity and apparent affinities) is clearly validated and again an interesting feature of the results, although somewhat expected on the basis of previous work (that is cited).

Overall the MS is very well presented and written.

I have a few relatively minor questions/ comments I believe the authors should address.

It wasn't clear to me how the "concentration" of the inositol phosphate tools once bound to the sepharose beads (incidentally I couldn't find the source of the streptavidin beads used) were calculated for the purposes of estimating apparent affinities. I assumed it was the amount known to be bound to the beads over the total aqueous volume with the beads. It is possible to argue it is more appropriate to use the volume of beads as that is space in which ligands are presented.

We thank the reviewer for this comment. The concentration refers to the amount of affinity reagent added relative to the total volume (volume of lysate/buffer + volume of beads; using a fixed amount of streptavidin beads). We agree that the local concentration of the immobilized molecules will be higher, but quantifying the bead volume can be cumbersome. We have clarified this point in the materials and methods section. We have also stated the source of the streptavidin beads now (and apologize for the oversight).

For bait immobilization we do not express a concentration per se. Instead, the amount of affinity reagent is defined relative to the volume of streptavidin used (i.e. 30 nmol of affinity reagent was incubated in in 80 µL of streptavidin). This can be found in the experimental section "Affinity enrichment for proteomics analysis and immunoblotting". These conditions were previously optimized to ensure efficient capture.

It would be helpful to know what typical proportions of proteins remained on the beads after elution with excess competing ligands.

We appreciate the reviewer's suggestion. We have now included this analysis in the Supporting Information section using immobilized InsP₆ and either recombinant protein XPR1^{SPX} or HEK293T cell lysates. To assess whether proteins remained bound to the beads after elution with excess competing ligand, the beads were subsequently subjected to chemical and thermal denaturing conditions to elute any remaining protein(s) (new Supporting Figure S6). For the recombinant protein, we could not observe any additionally eluted protein, and also for the cell lysate, the fraction removed under denaturing conditions was small (except for streptavidin). We conclude that elution with excess ligand seems to work quite well under our experimental conditions.

It would also be helpful to know how the modified, metabolically resistant pyro-phosphate analogues compared to the natural species in their ability to compete, giving an indication of the extent to which the modifications impacts binding.

We thank the reviewer for this comment. We have now addressed this point in the section "A proteomics workflow for determination of apparent binding constants" (lines 289-305). Specifically, we performed competition assays in which increasing concentrations of the non-modified, endogenous ligands were added prior to enrichment with the corresponding affinity reagents. These experiments were conducted for multiple ligands and representative targets, including PTK2, NAMPT, STK24, PRPS1, DIPP1, and PMVK, and were visualized by Western blot (New Figure 4b). Across targets, ligands, and affinity reagents, we observed clear and dose-dependent competition profiles, strongly suggesting that endogenous ligands and the corresponding non-hydrolysable affinity reagents compete for the same binding sites. While it is reasonable to assume that the derivatization will have some impact on the binding affinity of the reagents, the results indicate that the selected proteins interact with both naturally occurring, non-chemically modified inositol (pyro)phosphates as well as with their modified analogues to a similar extent.

In the text the authors refer to 11plex TMT labelling yet in the methods section in the critical commercial reagents section it appears a 10-plex was used. Was this a 10-plex combined with use of a separate 1-plex kit?

We appreciate the reviewer's question. In our work we have used the TMT 10plexTM kit (Cat# 90111) with the TMT11-131C label reagent extension (Cat# A37724). This amine-reactive Thermo Scientific TMT11-131C increases sample multiplexing capability from 10-plex to 11-plex, enabling even higher throughput in protein identification and quantitative analysis by tandem mass spectrometry (<https://www.thermofisher.com/order/catalog/product/A34808>). The study does have some very useful comparisons between tool compounds that afford a good view of the selectivity of binding, however, it would be very useful for the field and this study if a non-biological enantiomer of either 1PP (3PP) or 1,5PP was available. The chemistry

did yield some very nice enantiomerically pure intermediates that would support this approach but it would be an unreasonable request if the relevant enantiomers were not already or easily derived.

We agree with the reviewer that the “non-biological” enantiomers could provide an additional control for assessing binding specificity. As the reviewer points out, our synthetic approach would, in principle, allow us to pursue these affinity reagents. However, the syntheses of these corresponding enantiomeric PP-InsPs are technically demanding and time consuming and would require a very serious synthetic effort. We can certainly see our group (or maybe in collaboration with another synthetic group) pursuing these additional PP-InsP affinity reagents in the future, but for the current manuscript we wanted to focus on the PP-InsPs with known biological functions.

In my opinion the main issue with this paper for Nat Comms is whether the advance/resource it offers is significant enough to merit potential publication. For me, big advances for the field would for example include identification of new inositol poly/pyro-phosphate binding proteins and establishing they are subject to changes in properties associated with binding under physiological conditions. Especially if there was evidence the regulation was dynamic under physiological conditions. For example, that changed levels of inositol poly/pyro-phosphate changed the activity/distribution/abundance of a relevant binding protein.

We thank the reviewer for raising this important point regarding the significance and impact of our study. We feel strongly that our work constitutes a very valuable resource for the community, due to several reasons:

For the first time, chemical syntheses of affinity reagents for 1PP-InsP₅, and 1,5(PP)₂-InsP₄ (InsP₈) are reported. These new reagents are then used, alongside affinity reagents for closely related messengers 5PP-InsP₅ and InsP₆ in proteomics experiments. In these proteomics experiments, we utilized the affinity reagents over a wide concentration range, which allowed us to determine apparent binding constants (K_D^{app}) for the different ligands, under different conditions (with and without Mg²⁺ ions). We subsequently validated several newly identified interactions biochemically, further corroborating the specificity and physiological relevance of our findings. To our knowledge, our data sets represent the first systematic quantification of protein-InsP₆/PP-InsP interactions at this level of resolution, providing a unique and valuable dataset for the community. We further characterized a somewhat unexpected – yet very notable – connection to proteins that would undergo PP-InsP mediated pyrophosphorylation. Using the proteomics data, we could identify previously uncharacterized modification sites.

While we agree with the reviewer that a full characterization of a dynamic physiological process by a specific PP-InsP, with temporal and spatial resolution, would be a wonderful outcome, we feel that these types of studies are now up to the researchers in the community.

In fact, we strongly hope that our data can be sourced for exactly these types of investigations, as different research groups already have their organism or cell line of choice established, alongside the relevant cell-based assays and read-outs. Nevertheless, we hope the reviewer appreciates, how difficult such an “optimal outcome” is to obtain. One current illustration of the challenges that are still associated with these investigations are the recent papers that are aiming to elucidate the mechanism by which PP-InsPs (more specifically InsP₈) control phosphate export via XPR1 (Xenotropic and polytropic retrovirus receptor 1): Many high profile papers have been published – and yet the story keeps on growing (Science 2024, PMID: 39325866; Nature 2024, PMID: 39169184; Nat Commun 2025, PMID: 40140662; Nat Commun 2025, PMID: 40113814; Mol Cell 2025, PMID: 40858110, etc.)

Taken together, our novel affinity tools, the proteome-wide apparent binding constants, the biochemical validation of functional targets, the interesting connection to protein pyrophosphorylation, and the broadly applicable workflow constitute a significant technical, methodological and biological advance that we believe will be of lasting value to the field.